# Modelling the ability of mass drug administration to interrupt soil-transmitted helminth transmission: Community-based deworming in Kenya as a case study

Nyuk Sian Chong [1,2], Stacey R. Smith? [3]*, Marleen Werkman [2,4,5], Roy M. Anderson [2,4]

**1** Faculty of Ocean Engineering Technology and Informatics, Universiti Malaysia Terengganu, Terengganu, Malaysia, **2** London Centre for Neglected Tropical Disease Research, Department of Infectious Disease Epidemiology, St. Mary's Campus, Imperial College London, London, United Kingdom, **3** Department of Mathematics and Faculty of Medicine, The University of Ottawa, Ottawa, Ontario, Canada, **4** MRC Centre for Global Infectious Disease Analysis, School of Public Health, Imperial College London, **5** Julius Center for Health Sciences and Primary Care, University Medical Centre Utrecht, Utrecht, The Netherlands

* stacey.smith@uottawa.ca

**Data Availability Statement:** All relevant data are within the manuscript.

## Abstract

The World Health Organization has recommended the application of mass drug administration (MDA) in treating high prevalence neglected tropical diseases such as soil-transmitted helminths (STHs), schistosomiasis, lymphatic filariasis, onchocerciasis and trachoma. MDA—which is safe, effective and inexpensive—has been widely applied to eliminate or interrupt the transmission of STHs in particular and has been offered to people in endemic regions without requiring individual diagnosis. We propose two mathematical models to investigate the impact of MDA on the mean number of worms in both treated and untreated human subpopulations. By varying the efficacy of drugs, initial conditions of the models, coverage and frequency of MDA (both annual and biannual), we examine the dynamic behaviour of both models and the possibility of interruption of transmission. Both models predict that the interruption of transmission is possible if the drug efficacy is sufficiently high, but STH infection remains endemic if the drug efficacy is sufficiently low. In between these two critical values, the two models produce different predictions. By applying an additional round of biannual and annual MDA, we find that interruption of transmission is likely to happen in both cases with lower drug efficacy. In order to interrupt the transmission of STH or eliminate the infection efficiently and effectively, it is crucial to identify the appropriate efficacy of drug, coverage, frequency, timing and number of rounds of MDA.

## Author summary

We determine the best options for annual and biannual mass drug administration to control soil-transmitted helminths. An additional round of drugs can allow weaker drugs to be used. We apply the results to a community-based deworming project in Kenya.

**Funding:** NSC acknowledges support from the Ministry of Education of Malaysia and the Faculty of Ocean Engineering Technology and Informatics, Universiti Malaysia Terengganu. SS? is supported by an NSERC Discovery Grant and an NSERC Alliance Grant. RMA is grateful to the Bill and Melinda Gates Foundation for research grant support via the DeWorm3 (OPP1129535) award to the Natural History Museum in London (http://www.gatesfoundation.org/). NSC, MW and RMA acknowledge joint Centre funding from the UK Medical Research Council and Department for International Development. Funding for the TUMIKA project was received from the Bill & Melinda Gates Foundation, the Joint Global Health Trials Scheme of the Medical Research Council, the UK Department for International Development, the Wellcome Trust and the Children's Investment Fund Foundation. The funders had no role in study design, data collection and analysis, decision to publish, or preparation of the manuscript.

**Competing interests:** The authors have declared that no competing interests exist.

## Introduction

More than one billion people worldwide are affected by neglected tropical diseases (NTDs) every year [1]. High prevalence NTDs such as onchocerciasis, soil-transmitted helminths (STHs), schistosomiasis and lymphatic filariasis lock people into poverty even though they are amenable to periodic deworming via the use of anthelmintic medicines, also known as preventive chemotherapy (PC) [2–5]. The World Health Organization (WHO) [6] has introduced three strategies of applying PC in controlling STH infections:

1. Mass drug administration (MDA), where PC is applied to the whole population of an endemic region at regular intervals, regardless of the infectious status of an individual.

2. Targeted chemotherapy, where anthelmintic medicines are given to specific risk groups of people (e.g., specified by age and gender) at regular intervals, regardless of the infectious status of an individual.

3. Selective chemotherapy, where anthelmintic medicines are given to infected individuals or individuals suspected to be infected.

In 2015, over 1.5 billion of doses of PC were administered to almost 1 billion individuals for at least one of the targeted infections: lymphatic filariasis, onchocerciasis, schistosomiasis, STHs and trachoma [4]. MDA strategies are not only safe and effective but also cost-effective in controlling these diseases [7, 8]; it costs between US $0.30 and US $0.50 per person treated in most settings [4]. To ensure the aim of MDA can be achieved successfully, improvements in water, sanitation and hygiene (WASH) infrastructure, hygiene education, training (such as planning, coordination, capacity building and data management and analysis) and human resources are crucial, including Social and Behaviour Change Communication (SBCC). Volunteers and experienced staff are needed to ensure that the drug-distribution strategy and implementation of MDA are carried out efficiently and effectively [4, 9–13].

The WHO recommends periodic MDA once a year if the prevalence of STH infections in the community is more than 20%—or twice a year if the prevalence exceeds 50%—in order to reduce the morbidity by reducing the worm burden [14]. To date, MDA, SBCC, WASH and hygiene education are the control strategies that have been widely implemented to reduce the rate of morbidity, especially in an endemic region with high prevalence of STH infections [9–11, 13, 15–21]. In addition, MDA is usually carried out annually or biannually by delivering anthelmintic medicines to the at-risk population of an endemic area via school-based or community-based programmes. The WHO aims to eliminate and control STH infections through MDA strategies by covering at least 75% of preschool and school-age children [4, 6]. Nevertheless, different countries apply different types of strategies in conducting MDA programmes, depending on the local or national policies [2, 12, 20, 22–24]. The WHO recommends a single dose of albendazole (400mg) or mebendazole (500mg) in MDA to treat soil-transmitted helminthiasis [25, 26]. Albendazole and mebendazole are highly effective in treating *Ascaris lumbricoides* and hookworm infections, but they are not as effective in treating *Trichuris trichiura* infection. The combination of albendazole and ivermectin is recommended for MDA of *T. trichiura* [22, 24, 27–30].

By treating children in school, a higher treatment coverage can be achieved because school-aged children in untreated communities are the carriers of the highest burden of STHs and consequently suffer the greatest setbacks to growth, health and cognition during their development. Nevertheless, treating children alone does not always reduce the transmission significantly; this is especially the case for hookworm, as adults are the major drivers of transmission [31]. Although scaling up the treatment to the whole community will lead to significant reduction in transmission, the main issue is the cost of treatment, which depends on demography,

coverage level, the population that is going to be treated and the frequency and duration of treatment. If the interruption of transmission does not happen from treating children alone, treatment has to be continued indefinitely unless WASH programmes, health education and SBCC can change the underlying conditions [31, 32].

A handful of authors have modelled MDA strategies to interrupt STH transmission. Anderson *et al.* [33] investigated the possibility of eliminating soil-transmitted helminthiasis if MDA is the only control strategy that has been taken. They discovered that the STH infection persists if MDA is only targeting pre-school-aged and school-aged children, except for the case where the reproduction number is low ($R_0 \leq 2$) and transmission intensity is low to medium. However, by considering WASH intervention, the value of $R_0$ decreases significantly, and hence the high-transmission setting can be reduced to a medium or low one. In addition, their model showed that, with higher coverage of MDA, it is possible to interrupt the transmission of STHs if MDA is targeted at both children and adults. Clarke *et al.* [34] conducted a systematic review and meta-analysis (searching via MEDLINE, Embase and Web of Science for articles published on or before November 5, 2015) to compare the effect of mass (community-wide) and targeted (children only) strategies on STH prevalence in school-aged children. Both regression models and meta-analysis showed that the prevalence of STHs in children would be significantly reduced if mass MDA is performed compared to targeted MDA. Bronzan *et al.* [35] assessed the impact of community-based integrated MDA on schistosomiasis and STH prevalence in Togo. They observed that the prevalence of both schistosomiasis and STHs in children aged 6 to 9 are significantly reduced compared to the baseline. Moreover, they noticed that there is a resurgence of hookworm infection in children who are living in areas with high prevalence and who did not receive treatment in the past half year. Hence they suggested that areas with high prevalence should not only continue with the MDA strategy but also require environmental improvements such as improvements in WASH infrastructure and practices in order to interrupt the transmission of STHs. Truscott *et al.* [36] examined heterogeneity in transmission parameters that play a major role in hookworm infection using the baseline data obtained from the TUMIKIA study in Kenya (which was a large, randomised trial). This study showed that prevalence is related to the $R_0$ value in a nonlinear manner (due to the effect of density-dependent fecundity) and that there is a clear increasing linear trend in mean $R_0$ values versus mean egg count. Moreover, they observed that the prevalence depends highly on the degree of parasite aggregation. As a result, they suggested different MDA approaches should be carried out, especially when prevalence is low; when prevalence is sufficiently low, the high degree of parasite aggregation indicates that STHs may concentrate at the household level or in a group of people who consistently fail to comply with treatment or who consistently do not or cannot adopt improved WASH practices. Although MDA has been shown to be safe and effective in controlling STH infections, MDA alone has yet to be proven an effective long-term solution. Issues of potential anthelmintic resistance and donor fatigue in particular have raised concerns around long-term sustainability in controlling the transmission of STHs. Vaz Nery *et al.* [37] thus highlighted the need to study the application of WASH intervention and behaviour change to sustainably control long-term STH infection alongside the implementation of MDA in order to maintain low prevalence or achieve the global target of infection elimination.

Here, we would like to examine the impact of MDA on the mean number of worms in a human population and the possibility of interrupting the transmission of STHs if MDA is implemented. Two mathematical models are proposed in this study:

1. an impulsive mean-worm model to examine the effect of MDA on the mean number of worms in a human population of size *N*;

2. a modified form of the impulsive mean-worm model to examine the dynamics of the mean number of untreated worms due to lack of treatment or inefficacy of drug in the host population after the application of MDA.

This paper is organized as follows. First, we introduce our impulsive mathematical models and determine approximate analytical solutions. We then discuss the dynamics of the mean number of worms in host populations for community-based MDA of the TUMIKIA project in Kenya by applying both proposed models. We conclude with a discussion.

## Mathematical models

A set of nonlinear ordinary differential equations depicting the dynamics of the approximate mean number of worms $M(t)$ in a human population of size $N$ (i.e., the total number of worms in a human population divided by $N$) and the infectious reservoir $L(t)$ in the habitat of a human host at time $t$, proposed by Anderson and May [38], is defined as follows:

$$\begin{aligned}
\frac{dM}{dt} &= \beta L - (\mu + \mu_1)M \\
\frac{dL}{dt} &= \frac{\lambda}{2}\phi(M;k,z)f(M;k,z)M - \mu_0 L \equiv \frac{\lambda}{2}\mathcal{F}(M;k,z)M - \mu_0 L,
\end{aligned} \tag{1}$$

where the associated parameters are defined in Table 1.

Since the observed pattern of worm numbers per host is well described empirically by a negative binomial distribution [38], the density-dependent constraints on adult worm fecundity can be described by

$$f(M;k,z) = \left[1 + \frac{(1-z)M}{k}\right]^{-(k+1)},$$

and the mating probability of adult worms is given by

$$\phi(M;k,z) = 1 - \left[\frac{1 + \frac{(1-z)M}{k}}{1 + \frac{(2-z)M}{2k}}\right]^{k+1},$$

where $k$ is the clumping parameter of the negative binomial distribution (where the degree of worm clumping is measured inversely by $k$), $\gamma$ is the strength of density-dependent constraints, $z = e^{-\gamma}$ measures the strength of the density-dependent effects on adult worm fecundity and

**Table 1. Variables and parameters in model (1).**

| Variable/ Parameter | Description |
|---|---|
| $M(t)$ | Mean number of worms in a human population of size $N$ at time $t$ |
| $L(t)$ | The infectious reservoir in the habitat of human host at time $t$ |
| $\beta$ | The transmission rate between human and reservoir |
| $\mu_1$ | The human death rate |
| $\mu_0$ | The parasite mortality rate |
| $\mu$ | The worm death rate |
| $\lambda$ | The within-host rate of egg production by female worms |
| $k$ | The clumping parameter of the negative binomial distribution |
| $\gamma$ | The strength of density-dependent effects on fecundity |
| $z$ | $e^{-\gamma}$ |

$\mathcal{F}(M; k, z) \equiv \phi(M; k, z) f(M; k, z)$. For a negative binomial probability distribution of worms per host, the prevalence of infection, $y$, is defined as

$$y = 1 - \left(1 + \frac{M}{k}\right)^{-k}.$$

By taking into account the fact that the lifespan of the infectious reservoir (about one month or less) is much shorter than the adult worm in the host (1–2 years), we expect the dynamics of $L(t)$ will reach equilibrium much faster than the mean number of worms, $M(t)$. We can thus rewrite model (1) as

$$\frac{dM}{dt} = (\mu + \mu_1)[R_0\mathcal{F}(M) - 1]M, \tag{2}$$

where $R_0 = \frac{\beta\lambda}{2\mu_0(\mu+\mu_1)}$ is the reproduction number of this model in the absence of density-dependent effects in adult worm fecundity.

Clearly, $M = 0$ is an equilibrium point for model (2). Endemic equilibria, $M_{eq}$, of model (2) exist whenever we solve $R_0\mathcal{F}(M_{eq}) - 1 = 0$. The numerical solution of $M_{eq}$ (using bisection method) is depicted in Fig 1. The stable solution of $M_{eq}$ is denoted by the solid line, whereas the unstable solution of $M_{eq}$ is represented by the dashed line. To avoid confusion, we denote the stable endemic equilibrium by $M^*$ and the unstable endemic equilibrium by $M_*$. A bifurcation point, $M_{bp}$, exists in model (2) (denoted by the filled circle in Fig 1), defined as follows:

$$M_{bp} = \frac{k}{1-z}\left\{1 - \left[\frac{2-z}{2(1-z)}\right]^{\frac{k+1}{k+2}}\right\}^{-1}\left\{\left[\frac{2(1-z)}{2-z}\right]^{\frac{1}{k+2}} - 1\right\}.$$

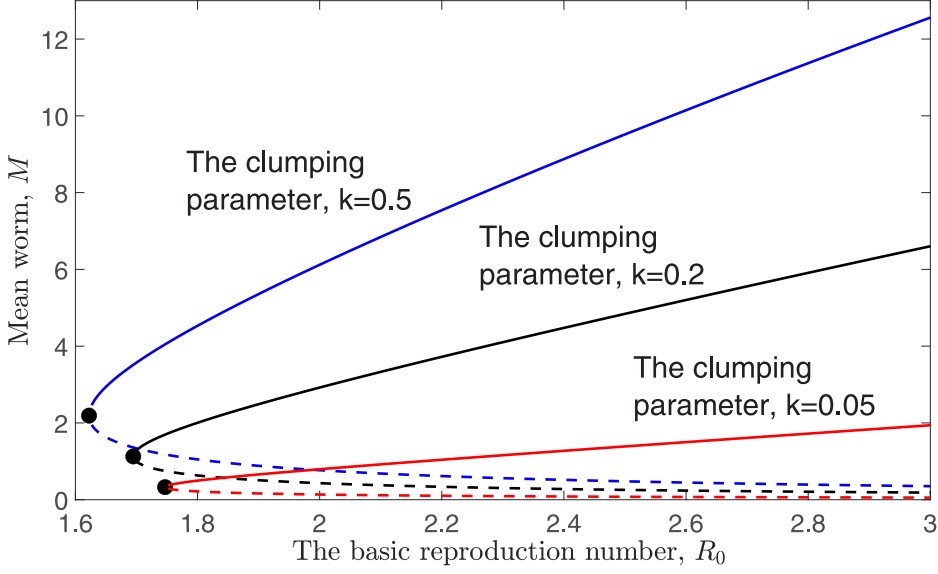

**Fig 1. Numerical solutions of the endemic equilibria $M_{eq}$ as a function of $R_0$ with different $k$ values, fixing $z = 0.96$.**

The corresponding reproduction number of $M_{bp}$ [38, 39] is given as follows:

$$R_{0bp} = \{\mathcal{F}(M_{bp})\}^{-1}.$$

The aggregation of STH parasites in the human host population is one of the key parameters in determining the prevalence of infection, $y$. Fig 1 shows that the endemic equilibrium and breakpoint of model (2) become smaller whenever $k$ is decreasing. In this case, we may have fewer individuals in the population who carry a higher burden of parasites if the probability distribution of STH parasites within the human host population becomes highly aggregated (i.e., when $k$ is small).

Next, we investigate the stability of the equilibrium point of model (2). An equilibrium occurs when there is no change in the rate of change of $M$, and the stability of an equilibrium point describes the effect of small perturbations near the equilibrium. If small deviations return to the equilibrium (or remain close), then the equilibrium is stable and is hence robust to noise and other random factors; if the equilibrium is unstable, then even very small perturbations will deviate from it over time. Since model (2) is one-dimensional, the stability of the equilibrium is determined by the sign of the derivative at the equilibrium.

**Theorem**. *Let $M_{eq}$ represent the endemic equilibrium of model (2). The disease-free equilibrium of model (2) is always locally asymptotically stable. The endemic equilibrium of model (2) is locally asymptotically stable if $M_{eq} > M_{bp}$ and unstable if $M_{eq} < M_{bp}$. A local bifurcation occurs at $M_{bp}$.*

**Proof**. Differentiating model (2) with respect to $M$, we have

$$\frac{\partial M'}{\partial M} = (\mu + \mu_1)[R_0 \mathcal{F}(M) - 1 + R_0 \mathcal{F}'(M)M].$$

In the absence of infection, $\frac{\partial M'}{\partial M} = -(\mu + \mu_1) < 0$ since all associated parameters are positive. Hence the disease-free equilibrium ($M = 0$) of model (2) is locally asymptotically stable. Moreover, the endemic equilibrium of model (2) is locally asymptotically stable whenever

$$(\mu + \mu_1)[R_0 \mathcal{F}(M_{eq}) - 1 + R_0 \mathcal{F}'(M_{eq})M_{eq}] \quad < 0$$

$$\frac{1-z}{k}\left\{1 - \left[\frac{2(1-z)}{2-z}\right]^{-\left(\frac{k+1}{k+2}\right)}\right\}M_{eq} \quad < \left[\frac{2(1-z)}{2-z}\right]^{\frac{1}{k+2}} - 1$$

$$M_{eq} > \frac{\left[\frac{2(1-z)}{2-z}\right]^{\frac{1}{k+2}} - 1}{\frac{1-z}{k}\left\{1 - \left[\frac{2(1-z)}{2-z}\right]^{-\left(\frac{k+1}{k+2}\right)}\right\}} \quad \equiv M_{bp},$$

where $\left[\frac{2(1-z)}{2-z}\right]^{\frac{1}{k+2}} > 1 \Rightarrow 1 - \left[\frac{2(1-z)}{2-z}\right]^{-\left(\frac{k+1}{k+2}\right)} < 0$, since all associated parameters are positive.

Therefore $M^*$ is locally asymptotically stable if $M_{eq} > M_{bp}$ (the infection will remain whenever the endemic equilibrium is larger than the breakpoint). Similarly, $M_*$ is unstable if $M_{eq} < M_{bp}$. Moreover, a local bifurcation occurs at $M_{bp}$, where $\frac{\partial M'}{\partial M}\big|_{M=M_{bp}} = 0$. This completes the proof.

Thus a separatrix exists, which splits the phase portrait of this model into two regions: trajectories that start above this separatrix tend to $M^*$, whereas trajectories with initial conditions below this separatrix will converge to $M = 0$.

## A standard mean-worm model with impulsive effect

WHO-recommended MDA is one of the most effective treatments to combat STH infection in endemic regions, and it is used to prevent morbidity caused by STH infection. By treating worms, the morbidity of STH can be reduced [15, 25, 38, 40–42]. We thus examine the impact of MDA on the dynamics of the mean number of worms in a human population of size $N$ by adding an impulsive effect into model (2). Hence our impulsive model, one that is governed by a set of impulsive differential equations, is defined as follows:

$$\frac{dM}{dt} = (\mu + \mu_1)[R_0 \mathcal{F}(M) - 1]M \quad t \neq t_n \tag{3}$$

$$M(t_n^+) = (1 - \omega\epsilon)M(t_n^-) \quad t = t_n, \tag{4}$$

where $t_n$ is the time when MDA is implemented, $n$ is an arbitrary positive integer, $\omega$ is the coverage of MDA and $\epsilon$ is the efficacy of drug. Impulsive differential equations generally lead to semi-continuous periodic orbits whose endpoints describe the local maxima and minima during each cycle. The endpoints of the impulsive orbit at time $t = t_n$ are $M(t_n^-)$ and $M(t_n^+)$, where $M(t_n^-)$ is the mean number of worms in a human population immediately before applying MDA, while $M(t_n^+)$ is the mean number of worms immediately after applying MDA.

Throughout this manuscript, we choose $z = 0.96$, $\mu = 0.5$ per year and $\mu_1 = \frac{1}{75}$ per year in performing the numerical simulations, unless otherwise stated. To examine the impact of MDA, numerical simulations that consider the application of one round of MDA at time $t = 1$ versus no treatment are compared, and the results are shown in Fig 2. For arbitrary $k$, $\omega$ and $\epsilon$

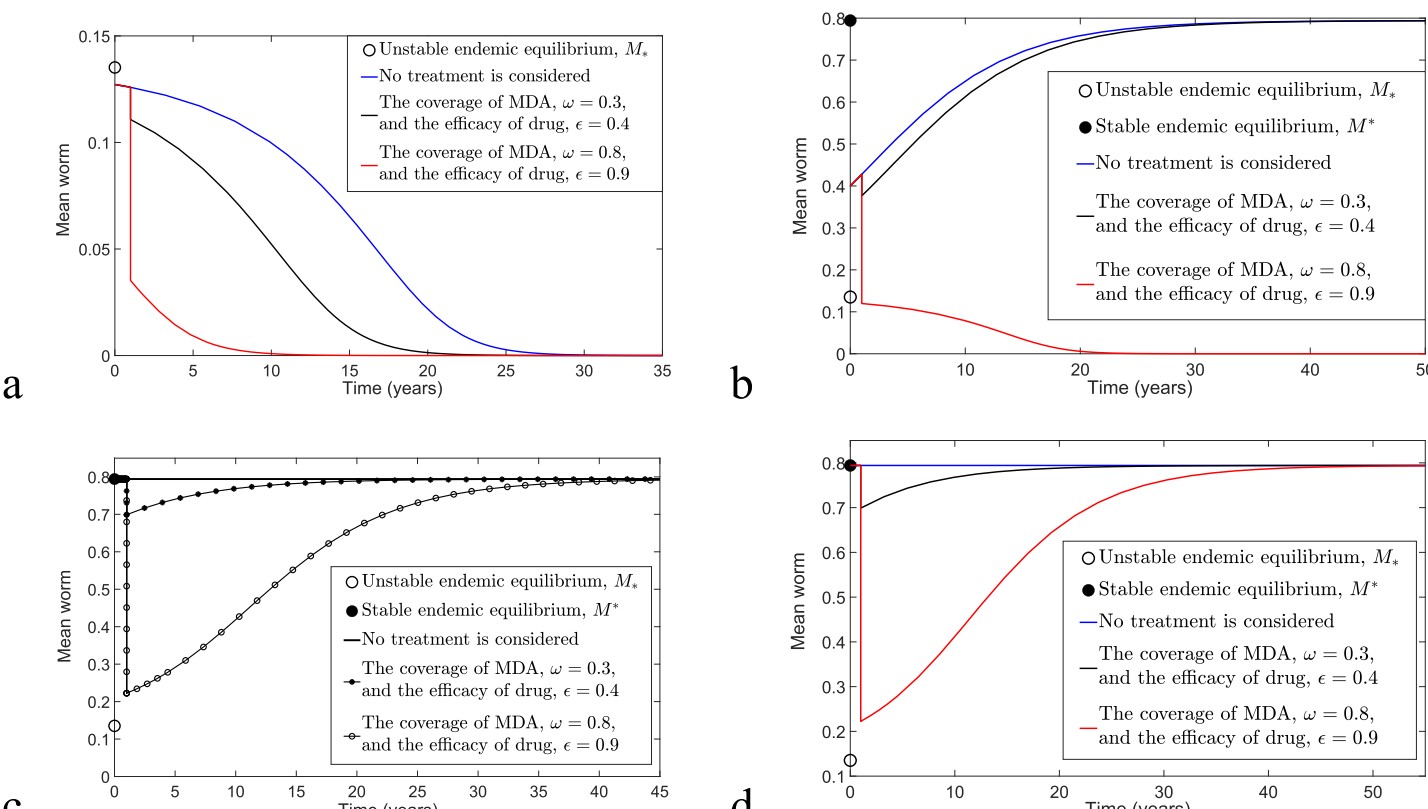

**Fig 2. Transmission dynamics of model (3) and (4) considering no application of MDA and one round of MDA at time $t = 1$ with $k = 0.05$ and $R_0 = 2$.** (a) $M_0 < M_*$. (b) $M_* < M_0 < M^*$. (c) $M_0 = M^*$. (d) $M_0 > M^*$.

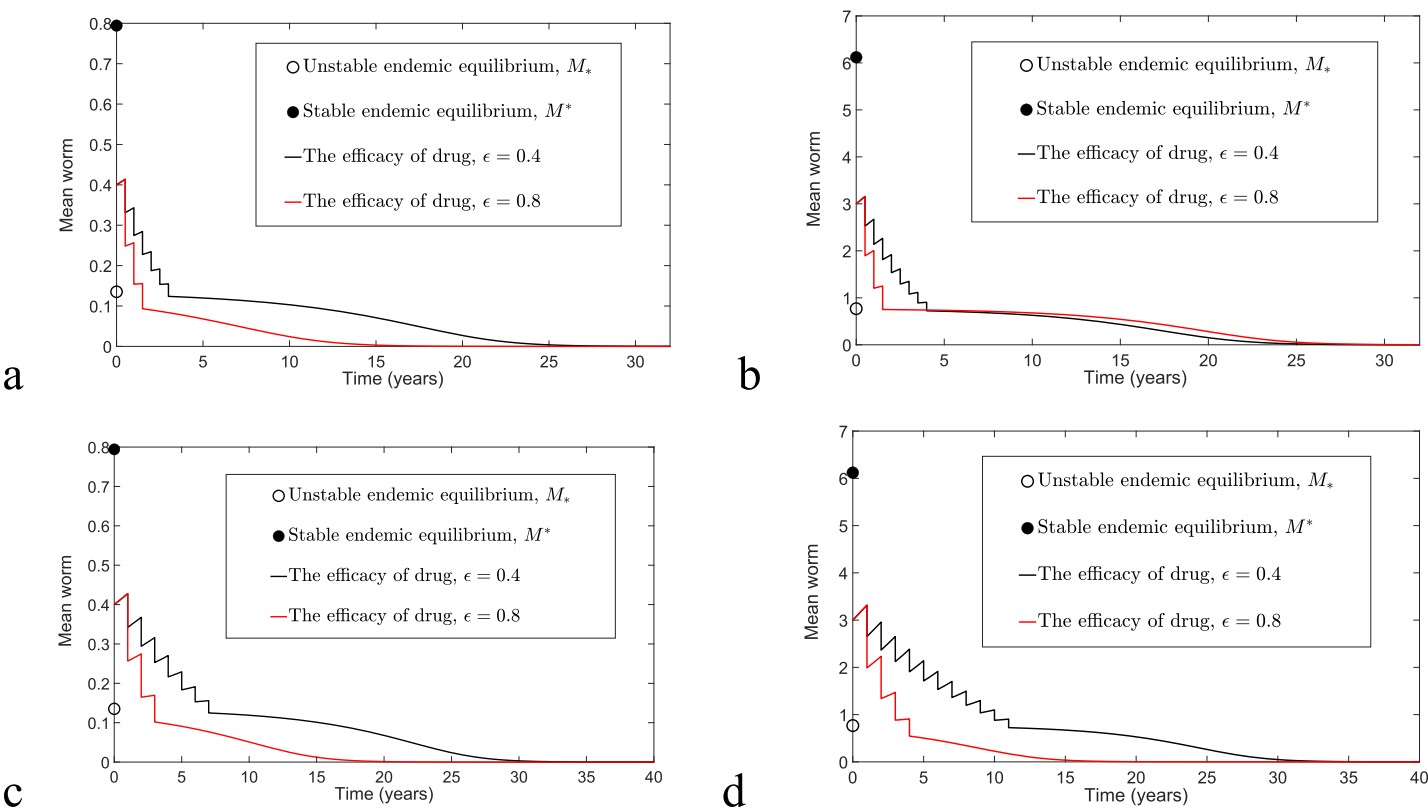

**Fig 3. Transmission dynamics of model (3) and (4) by considering the implementation of biannual and annual MDA, $M_0 > M_*$, varying $\epsilon$ and choosing $z = 0.96$, $R_0 = 2$ and $\omega = 0.5$.** Top row: biannual MDA. Bottom row: annual MDA. Left column: $k = 0.05$. Right column: $k = 0.5$.

values, all trajectories of model (3) and (4) as in Fig 2a and 2b are converging to zero if $M_0$, the initial value of $M$, is below the unstable endemic equilibrium $M_*$. However, with the application of one round of MDA at $t = 1$ (black and red solid curves), elimination of STH transmission can be achieved faster if $\omega$ and $\epsilon$ are sufficiently high compared to no treatment (blue solid curve). For sufficiently large $M_0$ (i.e., $M_0 = M^*$ as in Fig 2e and 2f), all solutions eventually approach the stable endemic equilibrium $M^*$, but trajectories with higher $\omega$ and $\epsilon$ values (red solid curves) take the longest time to reach $M^*$. This shows that the control strategy has reduced the mean number of worms and prolonged the time for the disease to achieve its endemic steady state. In addition, for arbitrary $M_* < M_0 < M^*$, eradication is possible if we could successfully interrupt the transmission (for instance, by applying treatment) and reduce the mean number of worms such that it is less than $M_*$ (see Fig 2c and 2d). Hence the solution of model (2) will converge to zero, and interruption of transmission is likely. Therefore, in order to achieve the target of transmission interruption, the application of the control strategy is crucial when the mean number of worms is greater than the transmission breakpoint, $M_*$.

Next, we examine the effect of applying biannual and annual MDA on STH transmission. By varying $\epsilon$ while selecting $\omega = 0.5$, $R_0 = 2$ and $M_* < M_0 < M^*$ in Fig 3, treatment is again unnecessary if $M < M_*$. From Fig 3, we observe that, for $k = 0.05$, six and three rounds of biannual MDA (respectively, seven and three rounds of annual MDA) are needed in order to eliminate STH infection if the efficacies of drug are 0.4 (i.e., 40% of the mean number of worms in the human population are removed if MDA is taken) and 0.8, respectively. Conversely, for $k = 0.5$, eight and three rounds of biannual MDA (respectively, eleven and four rounds of annual MDA) are required, with the aim of disease eradication, if $\epsilon = 0.4$ and 0.8, respectively.

These results show that more rounds of MDA are needed to eradicate the disease if $k$ is large, the drug efficacy is low and MDA is applied less frequently. The value of $M^*$ varies greatly if $k$ and $R_0$ values are increasing, but the value of $M_*$ does not vary much in this case, as shown in Fig 1.

**Overestimates.** We have shown that the STH infection will eventually decline (even without treatment) if $M < M_*$. Thus we examine the case $M_0 > M_*$ by using an overestimate of model (3) and (4) to estimate the number of rounds of MDA required to reduce the mean number of worms to less than the transmission breakpoint and hence interrupt STH transmission.

For arbitrary $M(t) > M_*$ and $t \neq t_n$ such that $t \in (t_{n-1}^+, t_n^-)$, Eq (3) is bounded by

$$\frac{dM}{dt} \leq (\mu + \mu_1)(R_0 \mathcal{F}_{\max} - 1)M, \text{ where } \mathcal{F}_{\max} = \mathcal{F}(M_{bp}).$$

Hence, for arbitrary $t \in (t_{n-1}^+, t_n^-)$ and $M(t) > M_*$, the solution of (3) is bounded by

$$M(t_n^-) \leq M(t_{n-1}^+)e^{(\mu+\mu_1)(R_0\mathcal{F}_{\max}-1)\Delta t}, \tag{5}$$

where the mean number of worms in a human population of density $N$ decays exponentially if $R_0 \mathcal{F}_{\max} < 1$ (respectively, grows if $R_0 \mathcal{F}_{\max} > 1$). After undergoing an impulse, the solution is bounded by

$$M(t_n^+) \leq (1 - \omega\epsilon)M(t_{n-1}^+)e^{(\mu+\mu_1)(R_0\mathcal{F}_{\max}-1)\Delta t}, \tag{6}$$

where $\Delta t = t_n - t_{n-1}$ is assumed fixed in our study since the MDA is usually applied at fixed intervals.

By considering $\mathcal{F}(M) \leq \mathcal{F}_{\max}$ for arbitrary $M$, we expect that the analytical solution (5) with (6) may overestimate the solution of model (3) and (4). Thus, to determine whether (5) and (6) provide a good approximation to the solution of model (3) and (4), we compare these two solutions for arbitrary $M(t) > M_*$ by taking into consideration both biannual and annual MDA control strategies in Figs 4 and 5, respectively. Moreover, by choosing $\omega = 0.5$, $\epsilon = 0.4$ and varying $k$ and $R_0$ values, the mean squared error (MSE) of these two solutions can be used to identify the magnitude of differences between these two solutions. Given any two functions, $g_1(t_i)$ and $g_2(t_i)$, where $i = 0, 1, 2, \ldots, m - 1$, the MSE is defined as follows [43, 44]:

$$\text{MSE} = \frac{1}{m} \sum_{i=0}^{m-1} [g_2(t_i) - g_1(t_i)]^2.$$

To find the endpoints of an impulsive orbit, first let us suppose that $M(t_0^+) = M_0$. Then we obtain

$$
\begin{aligned}
M(t_1^-) &\leq M_0 e^{(\mu+\mu_1)(R_0\mathcal{F}_{\max}-1)\Delta t} \\
M(t_1^+) &\leq (1-\omega\epsilon)M_0 e^{(\mu+\mu_1)(R_0\mathcal{F}_{\max}-1)\Delta t} \\
M(t_2^-) &\leq (1-\omega\epsilon)M_0 e^{2(\mu+\mu_1)(R_0\mathcal{F}_{\max}-1)\Delta t} \\
M(t_2^+) &\leq (1-\omega\epsilon)^2 M_0 e^{2(\mu+\mu_1)(R_0\mathcal{F}_{\max}-1)\Delta t} \\
&\vdots \\
M(t_n^-) &\leq (1-\omega\epsilon)^{n-1} M_0 e^{n(\mu+\mu_1)(R_0\mathcal{F}_{\max}-1)\Delta t} \\
M(t_1^+) &\leq (1-\omega\epsilon)^n M_0 e^{n(\mu+\mu_1)(R_0\mathcal{F}_{\max}-1)\Delta t}.
\end{aligned}
$$

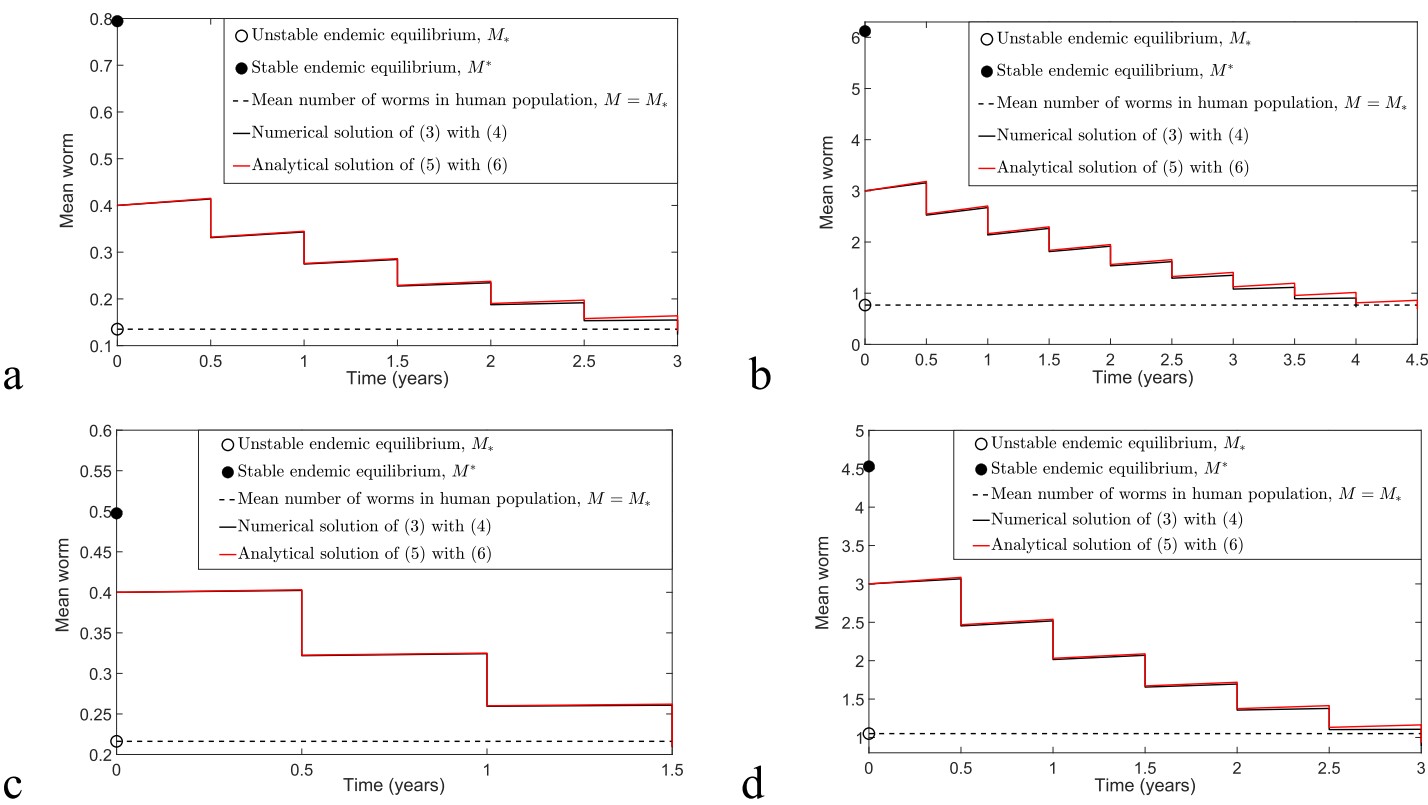

**Fig 4. The comparison of the overestimate (5) and (6) and original model (3) and (4) by considering the implementation of biannual MDA with $\epsilon = 0.4$ and $\omega = 0.5$ and varying $k$ and $R_0$ values.** (a) $R_0 = 2$, $k = 0.05$, $n > 5.83$ and MSE = $0.1258 \times 10^{-4}$. (b) $R_0 = 2$, $k = 0.5$, $n > 8.34$ and MSE = $0.2167 \times 10^{-2}$. (c) $R_0 = 1.8$, $k = 0.05$, $n > 2.86$ and MSE = $0.5217 \times 10^{-6}$. (d) $R_0 = 1.8$, $k = 0.5$, $n > 5.38$ and MSE = $0.6392 \times 10^{-3}$.

The impulsive orbit is bounded above by an orbit with endpoints

$$(1 - \omega\epsilon)^{n-1} M_0 e^{n(\mu+\mu_1)(R_0 \mathcal{F}_{\max}-1)\Delta t} \quad \text{and} \quad (1 - \omega\epsilon)^{n} M_0 e^{n(\mu+\mu_1)(R_0 \mathcal{F}_{\max}-1)\Delta t}.$$

Knowing the endpoints, we are able to find the estimated number of rounds of MDA such that $M(t_n^+) < M_*$ with fixed $\omega$ and $\epsilon$; i.e.,

$$n > \frac{\ln\left(\frac{M_*}{M_0}\right)}{\ln\left(1 - \omega\epsilon\right) + (\mu + \mu_1)(R_0 \mathcal{F}_{\max} - 1)\Delta t}, \tag{7}$$

or with fixed $n$ (and $\epsilon$ or $\omega$), the estimated values of $\omega$ and $\epsilon$ such that disease elimination is possible are given by

$$\omega > \frac{1}{\epsilon}\left[1 - \left(\frac{M_*}{M_0}\right)^{\frac{1}{n}} e^{-(\mu+\mu_1)(R_0 \mathcal{F}_{\max}-1)\Delta t}\right]$$

and

$$\epsilon > \frac{1}{\omega}\left[1 - \left(\frac{M_*}{M_0}\right)^{\frac{1}{n}} e^{-(\mu+\mu_1)(R_0 \mathcal{F}_{\max}-1)\Delta t}\right],$$

respectively.

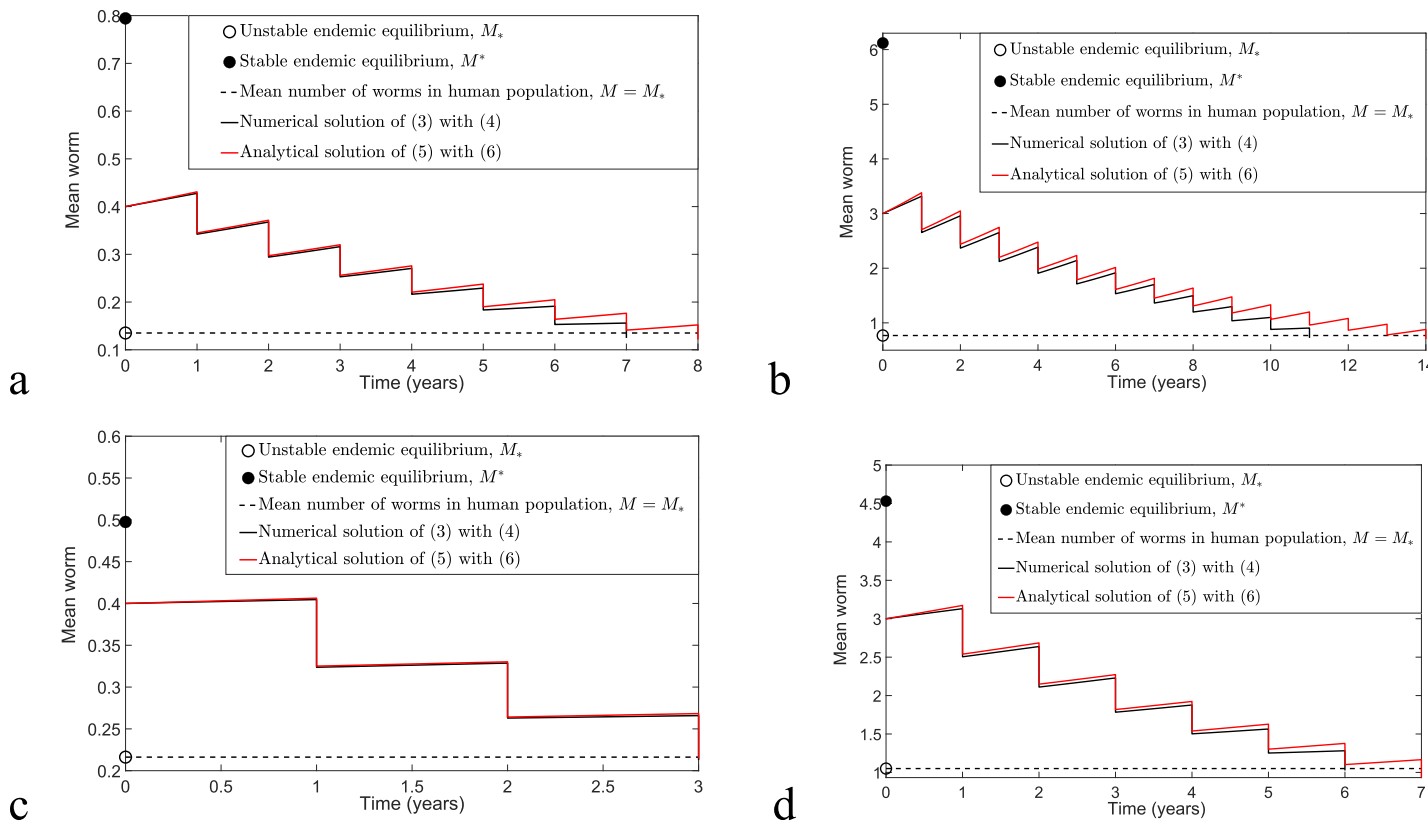

**Fig 5. The comparison of solutions (5) and (6) and numerical solutions of model (3) and (4) by considering the implementation of annual MDA with $\epsilon = 0.4$ and $\omega = 0.5$ and varying $k$ and $R_0$ values.** (a) $R_0 = 2$, $k = 0.05$, $n > 7.29$ and MSE $= 0.6333 \times 10^{-4}$. (b) $R_0 = 2$, $k = 0.5$, $n > 13.14$ and MSE $= 0.0156$. (c) $R_0 = 1.8$, $k = 0.05$, $n > 2.96$ and MSE $= 0.2106 \times 10^{-5}$. (d) $R_0 = 1.8$, $k = 0.5$, $n > 6.29$ and MSE $= 0.2264 \times 10^{-2}$.

From Figs 4 and 5, we discover that numerical solutions of both the overestimate (5) and (6) and the original model (3) and (4) are compatible if $k$ and $R_0$ values are sufficiently small. Moreover, both solutions have a better agreement for the case study of biannual MDA compared to annual MDA, where we can see that the MSE values for the biannual MDA are smaller than the annual MDA for arbitrary initial conditions, $k$ and $R_0$ values. Using Eq (7), the estimated number of rounds of biannual MDA such that interruption of STH transmission is feasible is well-matched with the numerical prediction for arbitrary $k$ and $R_0$ values, but it only has good agreement if $k$ and $R_0$ values are sufficiently small for annual MDA.

## A modified form of the impulsive mean worm model

In this section, we examine the behaviour of the mean number of untreated worms due to no treatment as well as inefficacy of drug in the host population after the application of MDA. Suppose that the mean number of worms in a human population of size $N$ before the application of any control strategy is $M$. By applying MDA, the human population of size $N$ can be divided into two subpopulations: treated and untreated, denoted $N_t$ and $N_{nt}$, respectively.

Next, assume that there are $\omega N$ treated people and $(1 - \omega)N$ untreated people, where $\omega$ is the coverage of MDA, and the total human population is $N = N_t + N_{nt}$. Let $M_t$ represent the mean number of untreated worms due to inefficacy of drug in the treated population and $M_{nt}$ represent the mean number of worms in the untreated population. For $t \neq t_n$, the dynamics of

$M_{nt}(t)$, $M_t(t)$ and $L(t)$ can be described by the following differential equations:

$$\frac{dM_{nt}}{dt} = \beta L - (\mu + \mu_1)M_{nt} \qquad\qquad t \neq t_n$$

$$\frac{dM_t}{dt} = \beta L - (\mu + \mu_1)M_t \qquad\qquad t \neq t_n \qquad (8)$$

$$\frac{dL}{dt} = \frac{\lambda}{2}[(1-\omega)\mathcal{F}(M_{nt})M_{nt} + \omega\mathcal{F}(M_t)M_t] - \mu_0 L \quad t \neq t_n.$$

Since the lifespan of adult worms in human hosts is much longer than the lifespan of larvae or eggs, the rate of change of $L(t)$ is expected to move much faster than $M_{nt}$ and $M_t$; hence we can rewrite model (8) using the equilibrium state of $L$ as follows:

$$\frac{dM_{nt}}{dt} = (\mu + \mu_1)\{R_0[(1-\omega)\mathcal{F}(M_{nt})M_{nt} + \omega\mathcal{F}(M_t)M_t] - M_{nt}\} \qquad t \neq t_n$$

$$\frac{dM_t}{dt} = (\mu + \mu_1)\{R_0[(1-\omega)\mathcal{F}(M_{nt})M_{nt} + \omega\mathcal{F}(M_t)M_t] - M_t\} \qquad t \neq t_n,$$

$(9)$

with impulse conditions

$$M_{nt}(t_n^+) = M_{nt}(t_n^-) \qquad\qquad t = t_n$$

$$M_t(t_n^+) = (1-\epsilon)M_t(t_n^-) \qquad\qquad t = t_n.$$

$(10)$

The disease-free equilibrium for model (9) is $(M_{nt}, M_t) = (0, 0)$, and the endemic equilibrium is $\widehat{M} = (M_{nt}, M_t) = (M_{nt}^*, M_{nt}^*) = (M_t^*, M_t^*)$, since $M_{nt}^* = M_t^*$. $M_t^*$ can be found by solving $R_0\mathcal{F}(M_t^*) - 1 = 0$ numerically, similar to the result obtained in Fig 1.

**Theorem**. *The disease-free equilibrium of model (9) is always locally asymptotically stable. The endemic equilibrium of model (9) is locally asymptotically stable if $R_0 < \frac{1}{\mathcal{F}(M_t^*)+M_t^*\mathcal{F}'(M_t^*)}$ and unstable if $R_0 > \frac{1}{\mathcal{F}(M_t^*)+M_t^*\mathcal{F}'(M_t^*)}$.*

**Proof**. Let $\widehat{\lambda}$ represent the eigenvalue of model (9). Then the characteristic equation is

$$\widehat{\lambda}^2 + (\mu + \mu_1)\{2 - R_0(1-\omega)[\mathcal{F}(M_{nt}) + M_{nt}\mathcal{F}'(M_{nt})] - R_0\omega[\mathcal{F}(M_t) + M_t\mathcal{F}'(M_t)]\}\widehat{\lambda}$$

$$+(\mu + \mu_1)^2\{1 - R_0(1-\omega)[\mathcal{F}(M_{nt}) + M_{nt}\mathcal{F}'(M_{nt})] - R_0\omega[\mathcal{F}(M_t) + M_t\mathcal{F}'(M_t)]\} = 0.$$

In the absence of infection, $\widehat{\lambda}_0 = -(\mu + \mu_1) < 0$ with multiplicity 2, since all associated parameters are positive. Hence the DFE is locally asymptotically stable. For the endemic equilibrium, $\widehat{M}$, we have

$$\widehat{\lambda}_1 = -(\mu + \mu_1) < 0 \quad\text{ and}$$

$$\widehat{\lambda}_2 = (\mu + \mu_1)\{R_0[\mathcal{F}(M_t^*) + M_t^*\mathcal{F}'(M_t^*)] - 1\} < 0 \quad\text{if } R_0 < \frac{1}{\mathcal{F}(M_t^*) + M_t^*\mathcal{F}'(M_t^*)}.$$

Therefore $\widehat{M}$ is locally asymptotically stable if $R_0 < \frac{1}{\mathcal{F}(M_t^*)+M_t^*\mathcal{F}'(M_t^*)}$. Otherwise, $\widehat{M}$ is unstable. This completes the proof.

To avoid confusion, we denote the stable endemic equilibrium by $\widehat{M}^*$, the unstable endemic equilibrium by $\widehat{M}_*$ and the initial value of $(M_{nt}, M_t)$ by $\widehat{M}_0$.

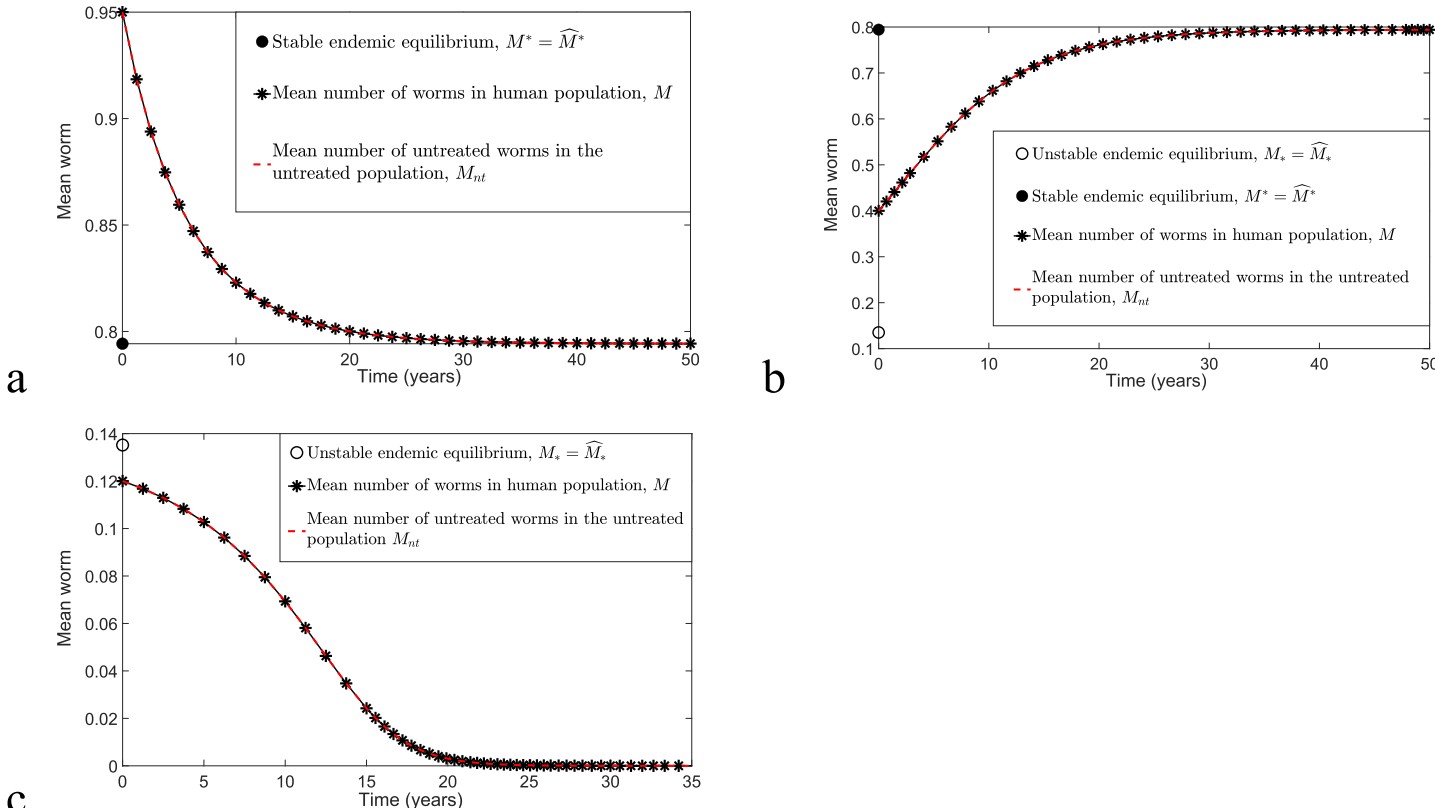

**Fig 6. The comparison of models (3) and (9) when no control strategy has been applied.** Both models have the same outcomes for arbitrary initial points, $R_0 = 2$ and $k = 0.05$. (a) $M_0 > M^*$ and $M_{nt0} > \widehat{M}_*$. (b) $M_* < M_0 < M^*$ and $\widehat{M}_* < M_{nt0} < \widehat{M}^*$. (c) $M_0 < M_*$ and $M_{nt0} < \widehat{M}_*$.

When there is no implementation of any control strategy, we have $N_t = 0$ and $N = N_{nt}$. Thus the transmission dynamics of $M_{nt}$ (model (9)) is similar to $M$ (model (3)), which is illustrated in Fig 6 for arbitrary initial conditions. STH infection remains endemic whenever the initial values of $M$ and $M_{nt}$ are greater than the unstable equilibrium $M_*$ or $\widehat{M}_*$ (in this case, these two models have similar equilibrium points; i.e., $M_* = \widehat{M}_*$ and $M^* = \widehat{M}^*$); see Fig 6a and 6b. Conversely, disease elimination is always possible if $M_0$ and $M_{nt0}$ are below the unstable equilibrium (see Fig 6c). We can thus infer that control strategies may not be necessary if $M(t) < M_*$ or $M_{nt}(t) < \widehat{M}_*$.

To investigate the impact of MDA in disease transmission, the transmission dynamics of $M_{nt}$ and $M_t$ with and without treatment are compared. By selecting $k = 0.05$, $R_0 = 2$ and the implementation of MDA at $t = 1$, the transmission dynamics of model (9) with impulse (10) for arbitrary $\widehat{M}_0 > \widehat{M}_*$, $\omega$ and $\epsilon$ is depicted in Fig 7. There is a decrease in the mean number of worms in both treated and untreated human subpopulations if treatment is taken compared to no application of MDA. However, if $M_{nt}$ and $M_t$ are still sufficiently large after treatment, this will increase the likelihood of disease spread, and hence STH infection will persist and remain endemic (see Fig 7a and 7b). The effects of $\omega$ and $\epsilon$ are more pronounced if $\widehat{M}_0$ is close to the neighbourhood of the unstable endemic equilibrium, $\widehat{M}_*$, in particular. From Fig 7c, we observe that disease elimination is likely to happen whenever $\omega$ and/or $\epsilon$ are sufficiently high, but STH infection persists if no control strategy is taken or if $\omega$ and $\epsilon$ are sufficiently low.

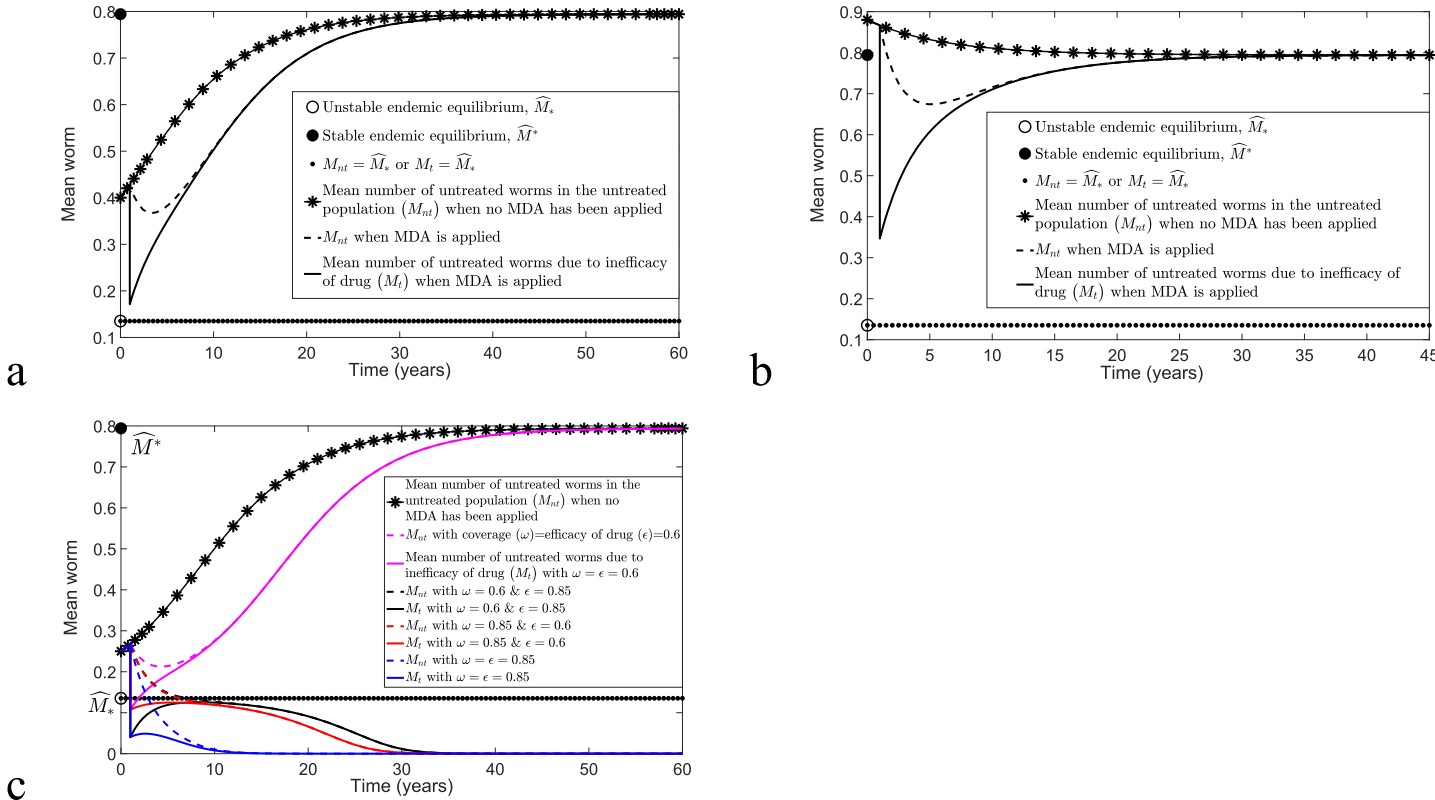

**Fig 7. The transmission dynamics of model (9) and (10) for a range of $\widehat{M}_0 > \widehat{M}_*$, $\epsilon$ and $\omega$ values, with $k = 0.05$ and $R_0 = 2$.** (a) $\omega = \epsilon = 0.6$ and $\widehat{M}_* < \widehat{M}_0 < \widehat{M}^*$. (b) $\omega = \epsilon = 0.6$ and $\widehat{M}_0 > \widehat{M}^*$. (c) $\widehat{M}_0$ around the neighbourhood of $\widehat{M}_*$.

Hence the application of MDA with appropriate $\omega$ and $\epsilon$ may hasten the elimination of STH infection.

**Approximate analytical solution of (9) and (10).**   In this subsection, we will focus on finding the approximate analytical solution of model (9) and (10) to estimate the required number of rounds of MDA in order to interrupt STH transmission for arbitrary $\widehat{M}_0 > \widehat{M}_*$.

Eq (9) is bounded by

$$\begin{aligned}
\frac{dM_{nt}}{dt} &\leq (\mu + \mu_1)\{[R_0(1-\omega)\mathcal{F}_{\max} - 1]M_{nt} + R_0\omega\mathcal{F}_{\max}M_t\} \\
\frac{dM_t}{dt} &\leq (\mu + \mu_1)[R_0(1-\omega)\mathcal{F}_{\max}M_{nt} + (R_0\omega\mathcal{F}_{\max} - 1)M_t].
\end{aligned} \tag{11}$$

Thus, for arbitrary $t \in (t_{n-1}^+, t_n^-)$, the analytical solution of (11) satisfies

$$\begin{aligned}
M_{nt}(t_n^-) &\leq [(1-\omega)M_{nt}(t_{n-1}^+) + \omega M_t(t_{n-1}^+)]e^{(\mu+\mu_1)(R_0\mathcal{F}_{\max}-1)\Delta t} \\
&\quad + \omega[M_{nt}(t_{n-1}^+) - M_t(t_{n-1}^+)]e^{-(\mu+\mu_1)\Delta t} \\
M_t(t_n^-) &\leq M_{nt}(t_{n-1}^+) + [M_t(t_{n-1}^+) - M_{nt}(t_{n-1}^+)]e^{-(\mu+\mu_1)\Delta t},
\end{aligned} \tag{12}$$

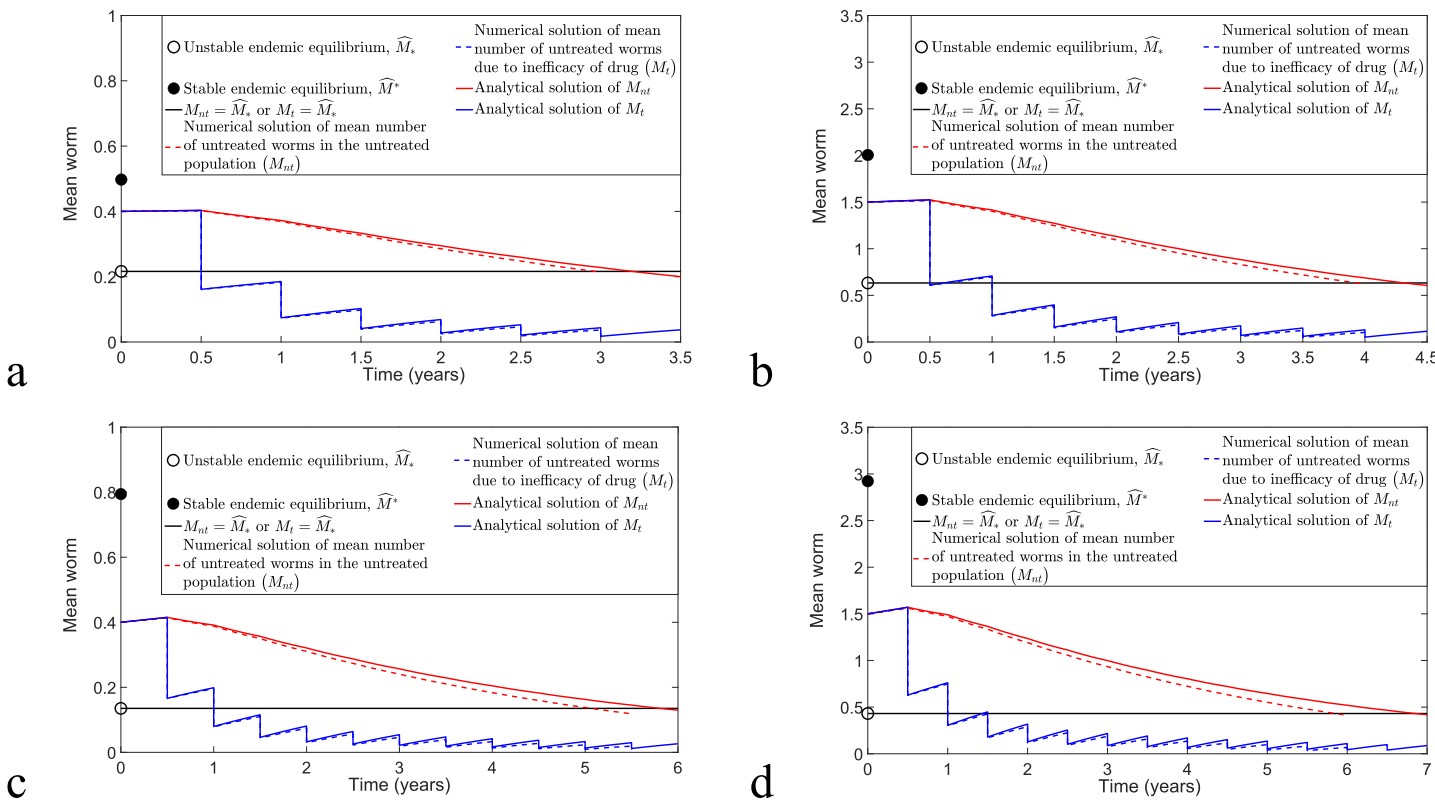

**Fig 8. Numerical comparison of model (12) and (13) and model (9) and (10) by considering the implementation of biannual MDA with $\epsilon = \omega = 0.6$ and varying $k$ and $R_0$ values.** (a) $k = 0.05$, $R_0 = 1.8$ and MSE for $M_{nt}$ and $M_t$ are $5.9377 \times 10^{-5}$ and $1.1881 \times 10^{-5}$, respectively. (b) $k = 0.2$, $R_0 = 1.8$ and MSE for $M_{nt}$ and $M_t$ are $1.5909 \times 10^{-3}$ and $2.2534 \times 10^{-4}$, respectively. (c) $k = 0.05$, $R_0 = 2$ and MSE for $M_{nt}$ and $M_t$ are $2.6274 \times 10^{-4}$ and $2.9925 \times 10^{-5}$, respectively. (d) $k = 0.2$, $R_0 = 2$ and MSE for $M_{nt}$ and $M_t$ are $4.5018 \times 10^{-3}$ and $4.7219 \times 10^{-4}$, respectively.

and, at impulse $t = t_n$, by applying (10), we have

$$
\begin{aligned}
M_{nt}(t_n^+) &\leq [(1-\omega)M_{nt}(t_{n-1}^+) + \omega M_t(t_{n-1}^+)]e^{(\mu+\mu_1)(R_0\mathcal{F}_{\max}-1)\Delta t} \\
&\quad + \omega[M_{nt}(t_{n-1}^+) - M_t(t_{n-1}^+)]e^{-(\mu+\mu_1)\Delta t} \\
M_t(t_n^+) &\leq (1-\epsilon)\{M_{nt}(t_{n-1}^+) + [M_t(t_{n-1}^+) - M_{nt}(t_{n-1}^+)]e^{-(\mu+\mu_1)\Delta t}\}.
\end{aligned}
\tag{13}
$$

We can compare numerical solutions of model (12) and (13) and model (9) and (10) with $\omega = \epsilon = 0.6$ and calculate the MSE to determine the magnitude of error between these two solutions. Fig 8 illustrates biannual MDA for both models, whereas Fig 9 compares annual MDA for these two solutions. By varying $k$ and $R_0$ values, the analytical and numerical solutions of $M_{nt}$ and $M_t$ in the case of biannual MDA demonstrate a better agreement compared to annual MDA. Moreover, we observe that analytical solutions forecast slightly higher values of $M_{nt}$ and $M_t$ and that more rounds of MDA are needed in order to achieve disease eradication compared to numerical solutions of model (9) and (10). This may be due to the approximations $\mathcal{F}(M_{nt}) \leq \mathcal{F}_{\max}$ and $\mathcal{F}(M_t) \leq \mathcal{F}_{\max}$ for arbitrary $M_{nt}$ and $M_t$. Nevertheless, both analytical and numerical solutions of $M_{nt}$ and $M_t$ lead to a good agreement if $k$ and $R_0$ are sufficiently small and there is a higher frequency of MDA (biannual MDA in our case).

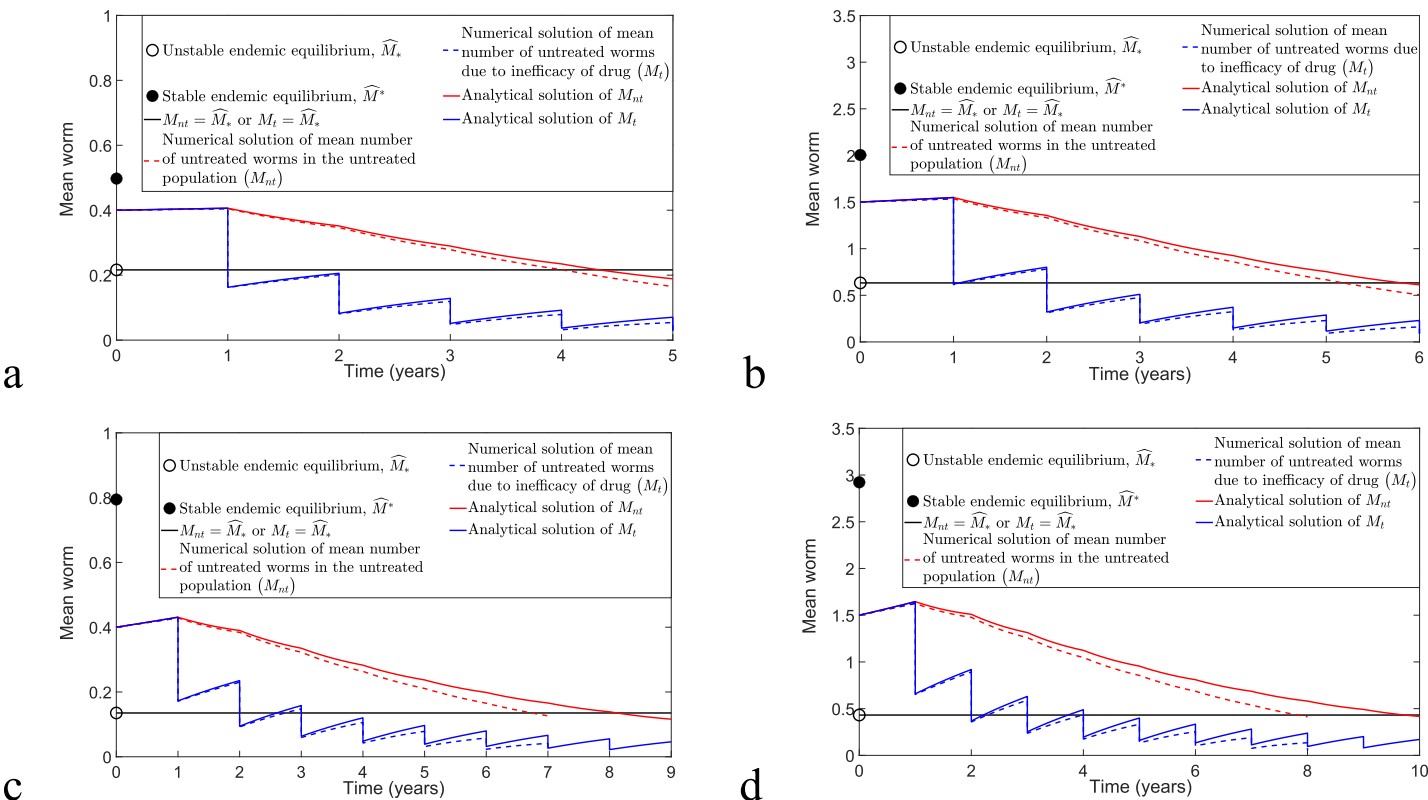

**Fig 9. Numerical comparison of model (12) and (13) and model (9) and (10) for annual MDA with $\epsilon = \omega = 0.6$ and varying $k$ and $R_0$ values.** (a) $k = 0.05$, $R_0 = 1.8$ and MSE for $M_{nt}$ and $M_t$ are $1.4241 \times 10^{-4}$ and $4.9371 \times 10^{-5}$, respectively. (b) $k = 0.2$, $R_0 = 1.8$ and MSE for $M_{nt}$ and $M_t$ are $3.2954 \times 10^{-3}$ and $9.5671 \times 10^{-4}$, respectively. (c) $k = 0.05$, $R_0 = 2$ and MSE for $M_{nt}$ and $M_t$ are $4.3739 \times 10^{-4}$ and $1.1943 \times 10^{-4}$, respectively. (d) $k = 0.2$, $R_0 = 2$ and MSE for $M_{nt}$ and $M_t$ are $8.6585 \times 10^{-3}$ and $2.1309 \times 10^{-3}$, respectively.

## Applications to the TUMIKIA project in Kenya

The Global Atlas of Helminth Infections [45] reported that approximately 15 million Kenyans are infected with STHs. The TUMIKIA project has the objective to assess which strategy is more efficient and effective in both controlling and eliminating soil-transmitted helminthiasis in Kenya: the combination of school- and community-based MDA versus school-based MDA alone. There were 120 community clusters in Kwale County, Kenya, that had been selected to participate in this project over two years, and 40 community clusters had been randomized to take part in one of the following three strategies [45–48]:

1. Annual school-based MDA programme, involving pre-school and school-aged children.

2. Annual community-based MDA programme, involving children and adults.

3. Biannual community-based MDA programme.

In this section, we focus on the impact of TUMIKIA community-based biannual and annual deworming strategies on the dynamics of $M$, $M_{nt}$ and $M_t$. A total of two (four) rounds of MDA had been implemented for a two-year community-based annual (biannual) MDA programme in Kenya. Moreover, the possibility of controlling STH infection in Kenya after the application of MDA for two years will be investigated. Model (3) and (4) will be employed to study the dynamics of $M$, whereas model (9) and (10) will be applied to study the dynamics of $M_{nt}$ and $M_t$. Since approximately 15 million Kenyans are infected with STHs [45], out of an

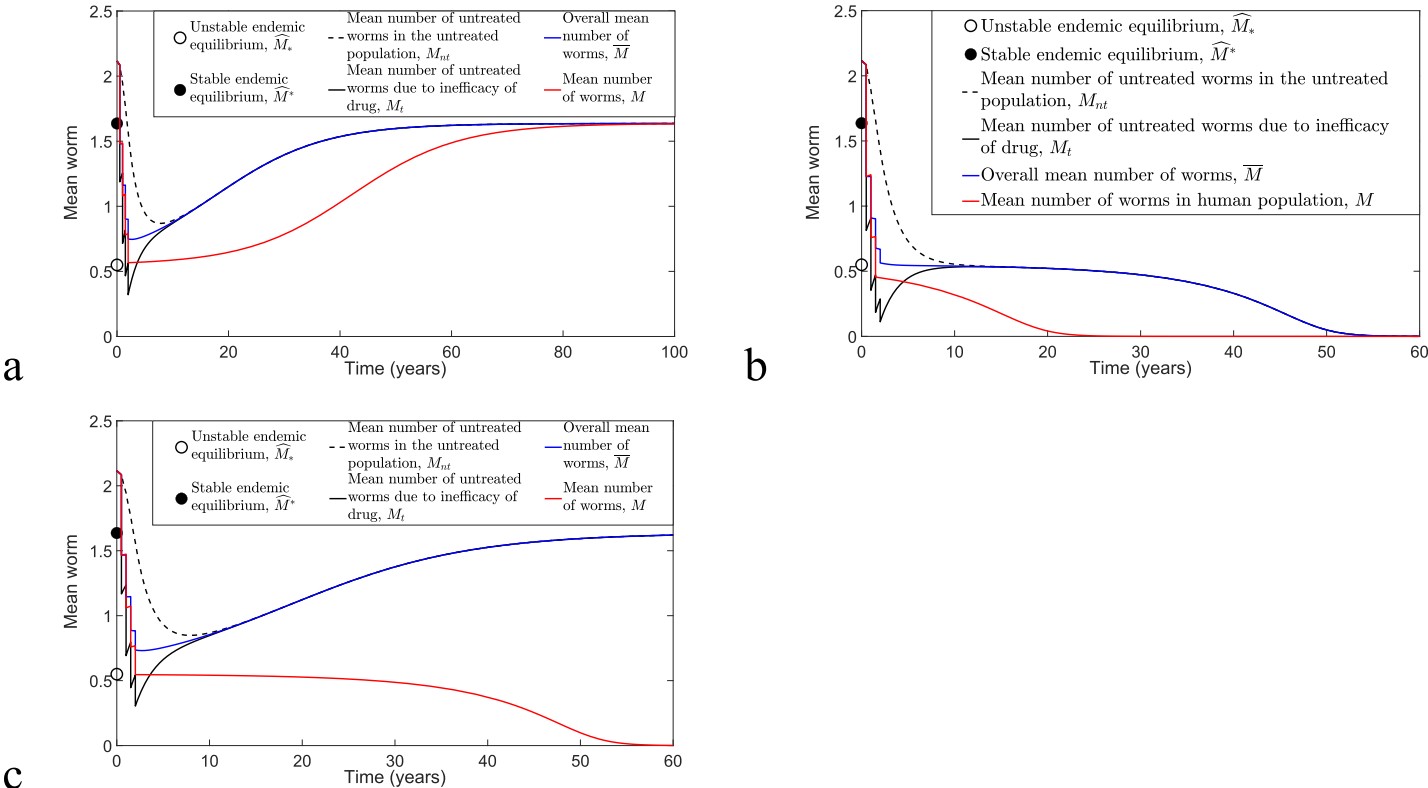

**Fig 10. Numerical results of model (3) and (4) and model (9) and (10) by varying $\epsilon$ and applying the coverage of MDA data from the TUMIKIA community-based biannual deworming control strategy.** (a) Disease persistence if $\epsilon < 0.44$. (b) Disease extinction is possible if $\epsilon \geq 0.61$. (c) Disease extinction is possible for model (3) and (4) if $\epsilon \geq 0.44$, but the disease will remain in endemic state for model (9) and (10) if $\epsilon < 0.61$.

estimated 43 million Kenyans [49], the initial prevalence value of STH infection in Kenya is about 0.3488. Thus, the estimated $k$ and $R_0$ values are 0.1624 and 1.8, respectively [36]. By taking the average of $\omega$ for all community clusters that had participated in each round of biannual and annual MDA programmes, the numerical simulations of model (3) and (4) and model (9) and (10) are demonstrated in Figs 10 and 11. The overall mean number of worms in both treated and untreated human populations is given by

$$\overline{M} = (1 - \omega)M_{nt} + \omega M_t.$$

From Fig 10, we see that, by implementing the TUMIKIA community-based biannual MDA strategy for two years, STH infection remains endemic for both models (3) and (4) and (9) and (10) if $\epsilon < 0.44$, but the infection dies off if $\epsilon \geq 0.61$ (see Fig 10a and 10b, respectively). However, Fig 10c shows that multiple outcomes are possible, depending on the model. That is, model (3) and (4) predicts interruption of transmission is possible if $\epsilon \geq 0.44$, but model (9) and (10) forecasts that STH infection persists if $\epsilon < 0.61$. By considering an additional round of TUMIKIA community-based biannual MDA at time $t = 2.5$ or $t = 7$, where the coverage of MDA is increased to 0.72, we notice that, in Fig 12, interruption of STH transmission is possible even though $\epsilon = 0.44$.

For the TUMIKIA community-based annual MDA strategy, we notice that in Fig 11a and 11b, both models predict disease persistence whenever $\epsilon < 0.73$ and disease extinction if $\epsilon \geq 0.85$, respectively. Nevertheless, we can see that there are multiple possible outcomes in

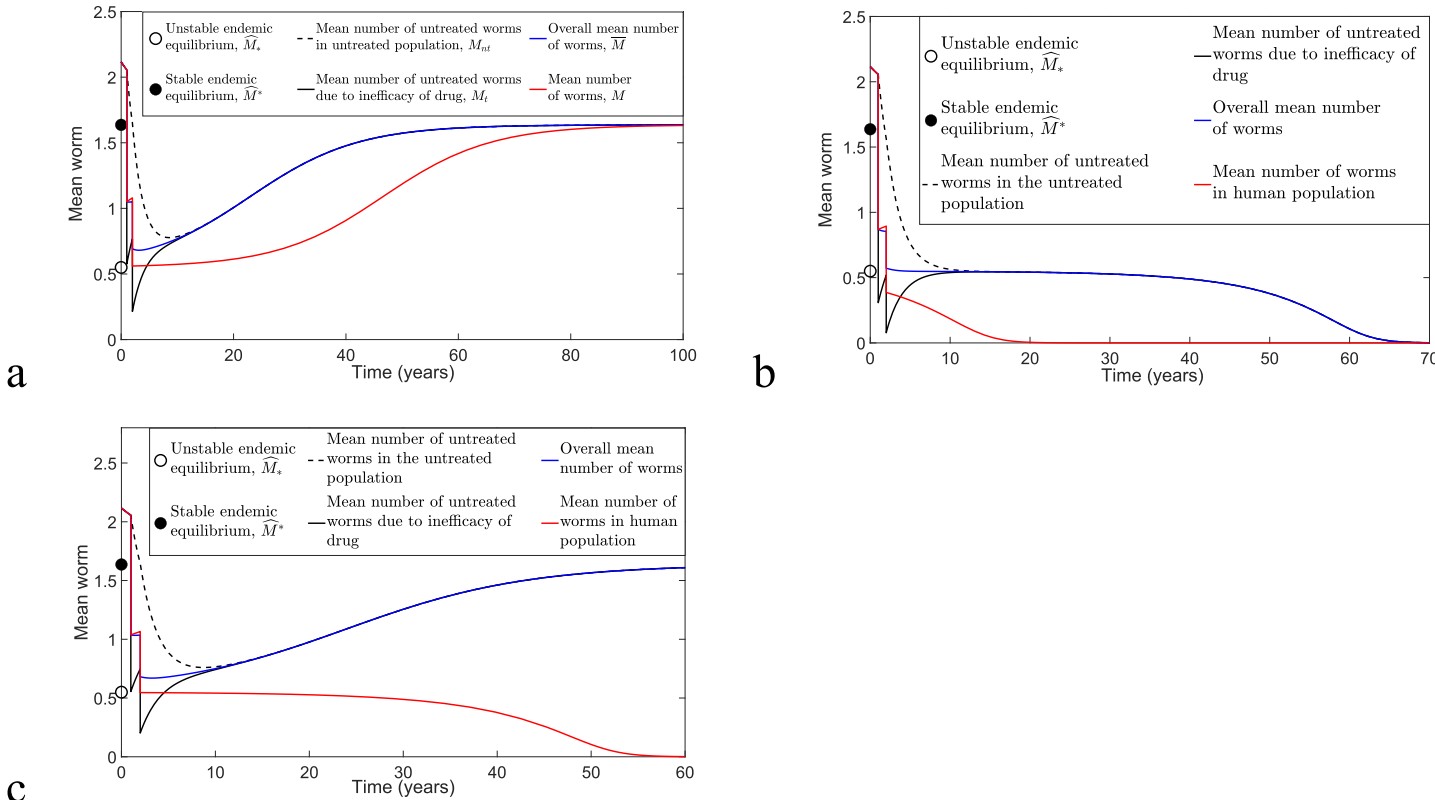

**Fig 11. The numerical results of model (3) and (4) and model (9) and (10) by varying $\epsilon$ and applying the coverage of MDA data from the TUMIKIA community-based annual deworming control strategy.** (a) Disease persistence if $\epsilon < 0.73$. (b) Disease extinction if $\epsilon \geq 0.85$. (c) Disease elimination is possible for model (3) and (4) if $\epsilon \geq 0.73$, but the disease will remain in endemic state for model (9) and (10) if $\epsilon < 0.85$.

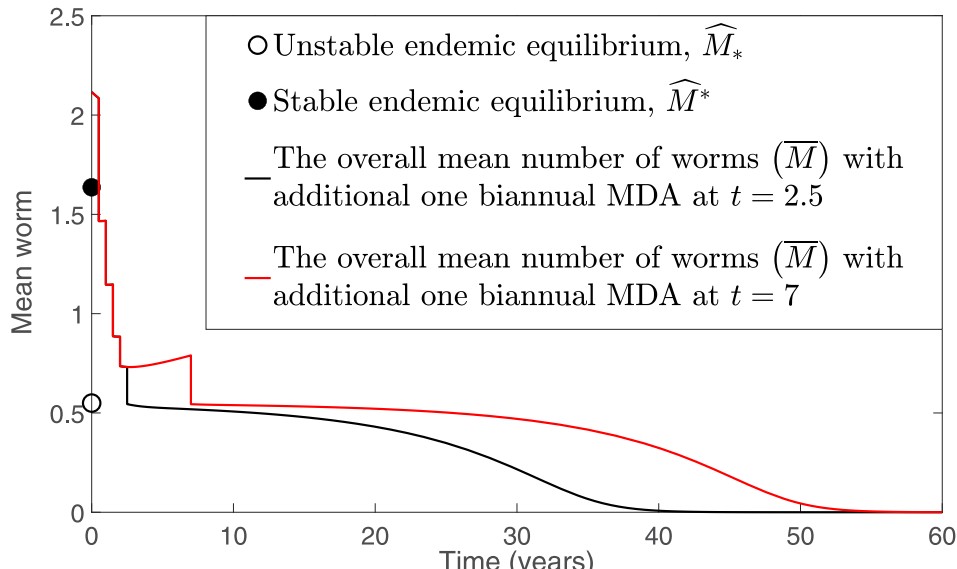

**Fig 12. Numerical solutions of model (9) and (10) with an additional round of TUMIKIA community-based biannual deworming strategy.**

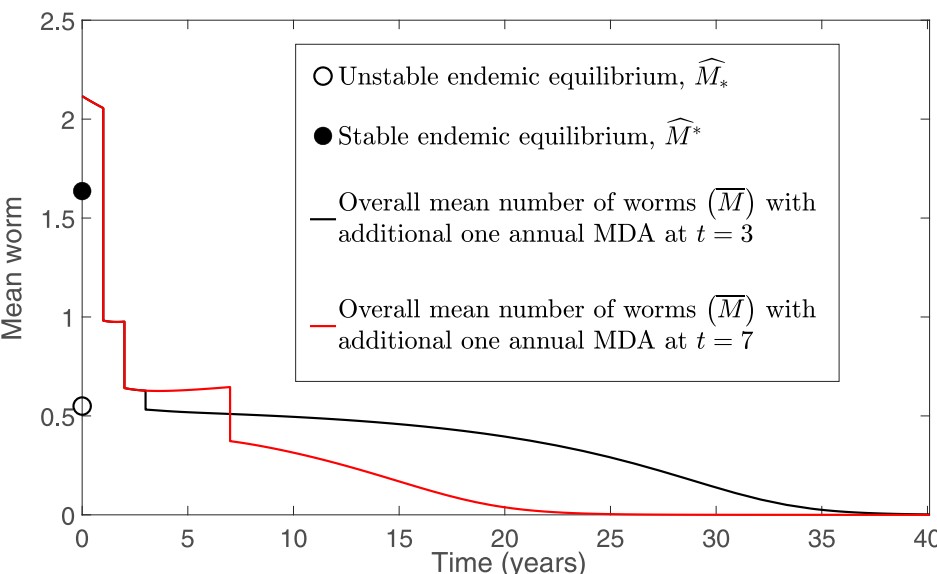

**Fig 13. Numerical solutions of model (9) and (10) with an additional round of TUMIKIA community-based annual deworming strategy.**

Fig 11c: model (3) and (4) forecasts disease eradication if $\epsilon \geq 0.73$, but model (9) and (10) predicts that disease eradication is unlikely if $\epsilon < 0.85$.

From Fig 11, we observe that a sufficiently high efficacy of drug is needed ($\epsilon \geq 0.85$) in order to eliminate STH infection for the TUMIKIA community-based annual MDA programme using model (9) and (10). Thus we investigated the possibility of transmission interruption by considering the application of an additional round of MDA. Two cases are examined: an additional round of MDA is employed in the third and seventh years after the last round of TUMIKIA community-based annual deworming (i.e., at times $t = 3$ and 7, respectively) with coverage $\omega = 0.6$. From Fig 13, we see that the STH infection in both cases will decline if $\epsilon \geq 0.77$. However, the infection is eliminated faster than in Fig 11. Since the rates of change of $M_{nt}$ and $M_t$ are very small around the unstable endemic equilibrium $\widehat{M}_*$, the interruption of STH infection in Fig 11 takes a long time to reach a disease-free state. Hence it is crucial for us to assess the appropriate coverage of MDA, efficacy of the drug and timing of the deworming strategy if we aim to achieve interruption of transmission quickly and effectively.

## Discussion

To examine the impact of MDA on the dynamics of STH infection in host populations and the feasibility of interrupting the transmission, two novel impulsive mathematical models were proposed: 1) a standard mean-worm model (3) and (4) to describe the effect of deworming strategy on the mean number of worms in a host population of size $N$, and 2) a modified form of the standard mean-worm model (9) and (10) to describe the dynamics of the mean number of worms in treated and untreated human subpopulations undergoing MDA.

Both models forecast that the application of MDA is unnecessary if the initial conditions are below the transmission breakpoint. If the initial number of worms is greater than the transmission breakpoint, then the mean number of worms in the host population can be suppressed using sufficient control strategies. Otherwise, there will be recrudescence in the community. Moreover, we found that fewer rounds of MDA are required to interrupt STH transmission if

the clumping parameter ($k$) and basic reproduction number ($R_0$) are sufficiently small or if the efficacy of drug, coverage and frequency of MDA are sufficiently high.

We also used both models to investigate worm dynamics for community-based MDA in the TUMIKIA project in Kenya. Both models predicted interruption of transmission was possible if the efficacy of drug was suffiently high (greater than 61% for the biannual programme or greater than 85% for the annual programme), whereas persistence was guaranteed if the efficacy was sufficiently low (less than 44% or 73%, respectively). The two models had different predictions when the efficacies were between these values. However, we showed that an additional round of MDA significantly improved the possibility of transmission interruption.

Our models have some limitations, which should be acknowledged. We set the infectious reservoir to steady state, due to the different timescales, although this may not apply perfectly. Impulsive differential equations assume the time-to-peak of MDA is negligible, which is only valid so long as the time between drug administrations is sufficiently large. STH eggs remain viable in soil for many years, so recrudescence of STH is inevitable without improvements in access to WASH and the adoption of new behaviours by the communities affected. We did not validate how long before STH recrudescence occurs if there is no WASH strategy or Social and Behaviour Change Communication. Finally, it is also crucial that the strategy implementation is respectful of the local context, traditional authorities, customs and belief systems.

Future work will consider age structure in human hosts, the effect of stochastic perturbations on disease transmission and assess the effectiveness of different potential interventions. We will also include a cost–benefit analysis of the MDA schedules. Implementing MDA programs takes time, coordination and costs, with volunteers increasingly expected to be paid for their services. More effective drugs delivered less frequently might be significantly cheaper for a health service than a cheaper drug requiring more rounds of MDA.

## Acknowledgments

The authors are grateful to James Truscott and Robert Hardwick for fruitful discussions. For citation purpses, please note that the question mark in "Smith?" is part of her name. We extend our sincere thanks to the researchers who led data collection in Kenya during the TUMIKIA trial, including Paul M Gichuki, Stella Kepha, Carlos Mcharo, Stefan Witek-McManus, Mary W Karanja, Leah Musyoka, Lennie Mutisya, Tuva Safari, Idris Muye and Maureen Sidigu. Scientific oversight during data collection was provided by Charles Mwandawiro, Sammy Njenga and Simon Brooker. We also thank the many study participants who consented to provide stool samples, and the large study team, including managers, fieldworkers, laboratory technicians, and drivers. TUMIKIA was conducted in collaboration with the Government of Kenya Ministry of Health (MoH) and the Kwale County MoH. The views, opinions, assumptions or any other information set out in this article are solely those of the authors.

## Author Contributions

**Conceptualization:** Nyuk Sian Chong, Stacey R. Smith?, Roy M. Anderson.

**Data curation:** Marleen Werkman.

**Formal analysis:** Nyuk Sian Chong.

**Funding acquisition:** Roy M. Anderson.

**Investigation:** Nyuk Sian Chong.

**Methodology:** Stacey R. Smith?.

**Software:** Nyuk Sian Chong.

**Supervision:** Roy M. Anderson.

**Writing – original draft:** Nyuk Sian Chong.

**Writing – review & editing:** Stacey R. Smith?, Marleen Werkman, Roy M. Anderson.

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
