## [Decision Letter · Decision Letter 0]

5 Mar 2021

Dear Dr. Smith?,

Thank you very much for submitting your manuscript "Modelling the ability of mass drug administration to interrupt soil-transmitted helminth transmission: community-based deworming in Kenya as a case study" for consideration at PLOS Neglected Tropical Diseases. As with all papers reviewed by the journal, your manuscript was reviewed by members of the editorial board and by several independent reviewers. In light of the reviews (below this email), we would like to invite the resubmission of a significantly-revised version that takes into account the reviewers' comments. 

We cannot make any decision about publication until we have seen the revised manuscript and your response to the reviewers' comments, especially please try to translate your results into public health sense so that more audience and understand your performance well. Your revised manuscript is also likely to be sent to reviewers for further evaluation.

Sincerely,

Guo-Jing Yang

Associate Editor

Banchob Sripa

Deputy Editor

Reviewer's Responses to Questions

**Key Review Criteria Required for Acceptance?**

**Methods**

-Are the objectives of the study clearly articulated with a clear testable hypothesis stated?

-Is the study design appropriate to address the stated objectives?

-Is the population clearly described and appropriate for the hypothesis being tested?

-Is the sample size sufficient to ensure adequate power to address the hypothesis being tested?

-Were correct statistical analysis used to support conclusions?

-Are there concerns about ethical or regulatory requirements being met?

Reviewer #1: Methods

I am unable to make any comment about the mathematical modelling, but if you are treating worms and of equal importance to public health across species it is flawed as 100 eggs of hookworm cause more morbidity than 100 of T. Trichuria v 100 of Acscaris.

Reviewer #2: (No Response)

Reviewer #3: I think my answer to the above questions is positive. By the way I dont have remarks about.

**Results**

-Does the analysis presented match the analysis plan?

-Are the results clearly and completely presented?

-Are the figures (Tables, Images) of sufficient quality for clarity?

Reviewer #1: Application to the Tumikia project in Kenya

L 269: Is this project aiming to control or to eliminate?

L 274: This sounds as if after 2 years you expect the problem to have been solved without potential recrudescence. Within 2 years you may get reduction or even control (<10% prevalence with <1% moderate/heavy infections) but recrudescence is inevitable of behaviours and WASH resources are not addressed no matter how many rounds of effective drugs are used.

Reviewer #2: (No Response)

Reviewer #3: Yes

**Conclusions**

-Are the conclusions supported by the data presented?

-Are the limitations of analysis clearly described?

-Do the authors discuss how these data can be helpful to advance our understanding of the topic under study?

-Is public health relevance addressed?

Reviewer #1: Discussion

L 335: You do not take into consideration that STH eggs remain viable in soil for many years, hence this interruption is only temporary.

L 341 Eradication is a completely new term and not appropriate here. Even elimination is inappropriate. I believe you are aiming for control whilst, WASH and human behavioral changes are strengthened.

Reviewer #2: (No Response)

Reviewer #3: Yes

**Editorial and Data Presentation Modifications?**

Reviewer #1: Author Summary 

L 2: I suggest 'control' rather than 'tackle' 

L 3: Can you make any comment regarding the cost of using an additional round of weaker drugs vs cost of using stronger drugs and fewer rounds?

Abstract 

Responsive to ‘preventive chemotherapy’. 

Is the mean number of worms the best way of perceiving the effort to reduce STH to ‘no longer of public health significant’ (<10% prevalence of any STH and <1% moderate/heavy infections) Do you actually mean worms of eggs per gram of faces?

There is such enormous variance in the morbidity of epg by species that you cannot compare 100epg for hookworm with 100epg for Ascaris: the morbidity is grossly different.

Reviewer #2: (No Response)

Reviewer #3: The paper can be published in the present form.

**Summary and General Comments**

Reviewer #1: Introduction

L 12: I suggest 'preventive chemotherapy (PC)' is more appropriate than 'deworming' as one doesn't know if the individual has actually been de-wormed the worm burden will have reduced but whether it gets to zero will depend upon mostly the worm burden before treatment as well as the type of medicine used (more v less effective)

L 18: I don't think 'only' is helpful here as targeted chemo and selective chemo and mass chemo often coexist in a community.

L 19: I don't think you need 'after a regular screening test'. Persons can self-refer for selective chemotherapy and be treated by clinicians/pharmacists without tests and/or buy over the counter medicines without consultations

L 21: Of doses of PC rather than 'treatments'

L 25: This brings be back to my comment about the costs of providing another round of MDA with a less effective agent versus the cost of less rounds of MDA with more effective agents.

L 26: PC should have been introduced earlier and then the abbreviation can be used throughout the manuscript.

L 30: I would add that the strategy needs to be respectful of the local context, traditional authorities, customs and belief systems if the last mile toward STH control is to be effective and that control to be maintained. I suggest that STH recrudesce is almost inevitable if these are not taken into consideration: if personal and environment hygiene are not considered.

L 33: This is a different definition to that used by the WHO 'no-longer of public health significance'

L 34: Important to also mention other important control strategies: improved water and sanitation (WASH).

L 38: Or community-based

L 41: Rather than use the word 'treating' I would use PC

L 44: Trichuris trichiura. The combination of ALB and IVM is recommended for LF-PC.

L 45: T. trichiura

L 48: Only strategy? No health education on WASH and/or efforts to improve safe water and improved sanitation?

L 53: Did this study continue to validate how long before STH recrudesce occurred if there truly was no strategy for WASH?

L 63: Areas with high baseline prevalence are especially in need of health education on personal hygiene and improved sanitation at household, community and school-level. MDA alone will be insufficient to eliminate and prevent resurgence. Rebound STH infection after mass MDA and apparent ‘control’ has been well documented for over 30 years.

L 73: Or consistently don't or cannot adopt improved WASH practices.

L 76: Again you could consider the cost-benefit of this approach, identifying monitoring and treating groups of individual versus improving WASH resources for a community/ or a vulnerable subgroup within a community.

Reviewer #2: The paper presents two mathematical models to guide strategy for MDA as an intervention for STH. The models have been thoroughly investigated, and forecasts for the mean number of worms in the population over time, for different drug efficacies, are presented. The authors investigated different levels of coverage. 

The paper does not suit a public health journal. The focus is on the mathematics, and the paper is written with inadequate translation to real life circumstances. The paper reads like an excellent mathematical exercise, but there is a lack of motivation that suits public health readers. For example, 

- What is the mean number of worms in a population? The mean of the estimate? The number of worms in a population is a scalar, not a distribution.

- Figure 1 shows the mean value of worms for different R0. There are three lines to correspond to three values of k, without any explanation as to why these values were chosen. The relation of k to the real life is lacking, i.e. it is impossible to interpret what different "clumping parameters of the negative binomial distribution" mean in the real world. What different settings would have a low/high k? 

- Why provide the eigenvalues of the models? What does the eigenvalue tell us? 

- Lines 116, "Moreover, the endemic equilibrium of model (2) is locally asymptotically stable whenever [an eqn where any relationships between parameters are too complicated to make the eqn easily interpretable] since all associated parameters are positive." What real world information does this tell us about the endemic equilibrium? Similarly on line 216

- L 177 "To find the endpoints of an impulsive orbit..." What is an impulsive orbit? Why do we need to calculate them? 

- There is regular mention of choosing arbitrary parameter values. Why? Are real life values impossible to obtain? In which case, why? Because the don't relate to real life or because the data is difficult to obtain? 

- How is the drug efficacy interpreted? Is it assuming perfect adherence? Is it the clearance rate?

With regards to writing style - it is unclear what parts of the model are new and the authors contribution to the field.

The application to Kenya data is underwhelming. The prevalence is averaged over the whole country, making the application very broad. It is not shown methodically that only (b) and (c) strategies are considered in this paper. There is a lack of clarity with regards to how many MDA rounds are used in (b) compared to (c). Results are converted from decimals to percentages in an inconsistent manner. The plots are provided without explanation as to the meaning of \\hat{M} etc (plots should be intepretable without having to read the paper). 

Abstract

- Don't include parameter notation in the abstract. 

- The acronym STH is introduced early on, and then throughout the paper the authors switch from writing out soil-transmitted helminthiasis in full, and using STH. 

Main text

- Line 38. The idea of focusing on children is alluded to, but not formally addressed. 

- Line 52 How low for R0?

- Line 53 Did Clark et al. [10] use data? This is unclear. 

- Line 64 Is MDA the deworming strategy?

- Line 68 states that there is variation in R0. In what sense? Under different settings? This statement doesn't provide enough information to make the reader feel that the work had a conclusion. 

- Line 81 to 86 Use the same wording where possible, so that the differences between the two models are immediately clear. Similarly for lines 275-280.

- Line 87-91 and line 212 are missing references to Sections. 

- Line 97 needs a reference to justify using a negative binomial distn. 

- Table 1 would benefit from z being added, even if stating exp(-\\gamma), to make the introduction to the model smoother. 

- The notation M_eq M^* M_* is unclear, and I'm not sure whether the authors interchange parts of these. In general, M^* and M_* are not distinctly different enough. 

- Line 125, n is a positive arbitrary integer. What are the units?

- Line 155 states that if k is large. From Fig 1 can you relate this to R0, and instead state that when R0 is large. This is a better reference point because R0 has real life meaning. 

- Line 176 What are these functions!? 

- For a public health journal, like the abstract, the discussion should avoid use of referring to parameters with their notation (i.e., k).

Reviewer #3: The problem that this manuscript takes in consideration is of great importance in some regions of the world. The authors carry on their analises with the help of some Mathematical models described by ODEs systems. Of special interest is the carefulness that the author put in the derivation of the form of constitutive functions. The numerical results are carefully compared with real data. The author discuss about some possible restrictions on the applicability of their study.The paper is clear and well organized, in my opinion it can be published in the present form.

PLOS authors have the option to publish the peer review history of their article (what does this mean?). If published, this will include your full peer review and any attached files.

Reviewer #1: No

Reviewer #2: No

Reviewer #3: No
---

## [Decision Letter · Decision Letter 1]

23 Jun 2021

Dear Dr. Smith?,

Thank you very much for submitting your manuscript "Modelling the ability of mass drug administration to interrupt soil-transmitted helminth transmission: community-based deworming in Kenya as a case study" for consideration at PLOS Neglected Tropical Diseases. As with all papers reviewed by the journal, your manuscript was reviewed by members of the editorial board and by several independent reviewers. The reviewers appreciated the attention to an important topic. Based on the reviews, we are likely to accept this manuscript for publication, providing that you modify the manuscript according to the review recommendations. 

Sincerely,

Guo-Jing Yang

Associate Editor

Banchob Sripa

Deputy Editor

Reviewer's Responses to Questions

**Key Review Criteria Required for Acceptance?**

**Methods**

-Are the objectives of the study clearly articulated with a clear testable hypothesis stated?

-Is the study design appropriate to address the stated objectives?

-Is the population clearly described and appropriate for the hypothesis being tested?

-Is the sample size sufficient to ensure adequate power to address the hypothesis being tested?

-Were correct statistical analysis used to support conclusions?

-Are there concerns about ethical or regulatory requirements being met?

Reviewer #1: Yes, objectives are clearly articulated and the study design is appropriate with clearly described population and hypothesis. the sample is sufficient and no concerns about ethical regulatory requirements

**Results**

-Does the analysis presented match the analysis plan?

-Are the results clearly and completely presented?

-Are the figures (Tables, Images) of sufficient quality for clarity?

Reviewer #1: Yes

**Conclusions**

-Are the conclusions supported by the data presented?

-Are the limitations of analysis clearly described?

-Do the authors discuss how these data can be helpful to advance our understanding of the topic under study?

-Is public health relevance addressed?

Reviewer #1: I think the imitations paragraph needs a little editing:

392: ‘so human-induced interruptions are only temporary; hence our results have less viability in the long term’ could you phrase this. Perhaps something like ‘recrudescence of STH is inevitable without improvements in access to WASH and the adoption of new behaviors by the communities affected’.

L394: and SBCC

L 396-398: could you also include a cost-benefit analysis of these various schedules of MDA. Implementing MDAs takes time, coordination and costs. Volunteers are increasingly expecting to be paid for their services so a more effective drugs delivered less frequently might be significantly cheaper for a health service than a cheaper drug requiring more rounds of MDA.

**Editorial and Data Presentation Modifications?**

Reviewer #1: (No Response)

**Summary and General Comments**

Reviewer #1: This is much improved and now fits better into the reality of programming in SSA. I still have a few comments:

Introduction

L22: ‘deworming’ use instead of ‘PC’ or MDA which is used more often going forward

L24-27: you have omitted the human-element: the change in behavior usually addressed by Social and Behavior Change Communication (SBCC)

L29: you have used PC here but almost everywhere else you have used MDA, please be consistent unless you are trying to differentiate between MDA and PC?

L31: ‘deworming’ 

L34: and SBCC

L39: now MDA is used instead of PC, please choose on or the other but don’t switch back and forth

L47: Trichuris trichiura has been introduced in full on L46 so it can be abbreviated to T. trichiura form then onwards

L48-49: Schools are not targeted because of cost-efficiencies but because the SAC are, in untreated communities the carriers of the highest burden of STHs and whilst growing suffer the greatest set-backs to growth, health and cognition. 

L56: and BCCC (which is different from just health education)

L58: MDA (or PC) but not ‘deworming’

L89: ‘deworming’

L98: presumable also of SBBC?

Discussion:

L373: ‘appropriate’ what is meant by that?

L375: ‘greater than’ or less than?

L375 ‘Otherwise, the infection may persist’ This sounds odd do you mean the infection in that individual or the risk of recrudescence within the community?

PLOS authors have the option to publish the peer review history of their article (what does this mean?). If published, this will include your full peer review and any attached files.

Reviewer #1: Yes: Dr Mary H, Hodges

Figure Files:

Data Requirements:

Reproducibility:

References

---

## [Decision Letter · Decision Letter 2]

5 Jul 2021

Dear Dr. Smith?,

We are pleased to inform you that your manuscript 'Modelling the ability of mass drug administration to interrupt soil-transmitted helminth transmission: community-based deworming in Kenya as a case study' has been provisionally accepted for publication in PLOS Neglected Tropical Diseases.

Best regards,

Guo-Jing Yang

Associate Editor

Banchob Sripa

Deputy Editor

Reviewer's Responses to Questions

**Key Review Criteria Required for Acceptance?**

**Methods**

-Are the objectives of the study clearly articulated with a clear testable hypothesis stated?

-Is the study design appropriate to address the stated objectives?

-Is the population clearly described and appropriate for the hypothesis being tested?

-Is the sample size sufficient to ensure adequate power to address the hypothesis being tested?

-Were correct statistical analysis used to support conclusions?

-Are there concerns about ethical or regulatory requirements being met?

Reviewer #1: (No Response)

**Results**

-Does the analysis presented match the analysis plan?

-Are the results clearly and completely presented?

-Are the figures (Tables, Images) of sufficient quality for clarity?

Reviewer #1: (No Response)

**Conclusions**

-Are the conclusions supported by the data presented?

-Are the limitations of analysis clearly described?

-Do the authors discuss how these data can be helpful to advance our understanding of the topic under study?

-Is public health relevance addressed?

Reviewer #1: (No Response)

**Editorial and Data Presentation Modifications?**

Reviewer #1: Line 82 and 369 deworming strategy still there (rather than MDA). In 369 especially it is potentially confusing as the second section specifically refers to MDA.

Line 392 you are now using the term drug administration rather than MDA

**Summary and General Comments**

Reviewer #1: (No Response)

PLOS authors have the option to publish the peer review history of their article (what does this mean?). If published, this will include your full peer review and any attached files.

Reviewer #1: **Yes: **Mary H Hodges

---

## [Editor Report · Acceptance letter]

28 Jul 2021

Dear Dr. Smith?,

We are delighted to inform you that your manuscript, "Modelling the ability of mass drug administration to interrupt soil-transmitted helminth transmission: community-based deworming in Kenya as a case study," has been formally accepted for publication in PLOS Neglected Tropical Diseases.

Best regards,

Shaden Kamhawi

co-Editor-in-Chief

Paul Brindley

co-Editor-in-Chief
